# GWAS for systemic sclerosis identifies six novel susceptibility loci including one in the Fcγ receptor region

Here we report the largest Asian genome-wide association study (GWAS) for systemic sclerosis performed to date, based on data from Japanese subjects and comprising of 1428 cases and 112,599 controls. The lead SNP is in the FCGR/FCRL region, which shows a penetrating association in the Asian population, while a complete linkage disequilibrium SNP, rs10917688, is found in a cis-regulatory element for IRF8. IRF8 is also a significant locus in European GWAS for systemic sclerosis, but rs10917688 only shows an association in the presence of the risk allele of IRF8 in the Japanese population. Further analysis shows that rs10917688 is marked with H3K4me1 in primary B cells. A meta-analysis with a European GWAS detects 30 additional significant loci. Polygenic risk scores constructed with the effect sizes of the meta-analysis suggest the potential portability of genetic associations beyond populations. Prioritizing the top 5% of SNPs of IRF8 binding sites in B cells improves the fitting of the polygenic risk scores, underscoring the roles of B cells and IRF8 in the development of systemic sclerosis. The results also suggest that systemic sclerosis shares a common genetic architecture across populations.

Systemic sclerosis (SSc) is one of the systemic autoimmune diseases (AIDs) characterized by fibrosis in connective tissues and internal organs, such as the lung or the kidney as a consequence of microvascular dysfunction and dysregulated immune systems[1]. Despite recent progress in disease management[2], SSc still has high morbidity and mortality mainly due to a poor understanding of its underlying pathophysiological mechanisms[1]. The etiology of SSc is complex and is not fully understood, but as with most AIDs, it is widely accepted that both environmental[3,4] and genetic factors contribute to the risk of the disease.

SSc can be classified according to clinical features or serological profiles. The clinical subtypes, limited cutaneous SSc (lcSSc) and diffuse cutaneous SSc (dcSSc), are based on the distribution of skin fibrosis. Organs involved in each subtype also differ and dcSSc tends to involve critical organs (e.g., the lungs), and hence has a worse prognosis than lcSSc. Anti-centromere antibody (ACA), and anti-topoisomerase antibody (ATA) are the two most prevalent autoantibodies and they do not usually co-exist in the same individual.

Importantly, the presence of the autoantibody often parallels with clinical subtypes and thus can be considered biomarkers for SSc. Indeed, ACA is more prevalent among lcSSc patients while ATA is more found in dcSSc subjects[5]. As observed in many polygenic diseases, ethnic differences might exist in SSc. It was reported that SSc in Asians is characterized by a younger age of onset, higher frequency of diffuse skin involvement, higher frequency of ATA and anti-U1 RNP antibodies, and more clinical phenotype leading to poorer prognosis compared to non-Asian SSc[6], implying the specific genetic architecture in a given ethnic group.

After the introduction of genome-wide association studies (GWASs)[7], the number of genetic markers convincingly associated with SSc has exponentially increased. The first GWAS for SSc was reported by Radstake et al in 2010[8] and there have been five large-scale GWASs published thereafter[9–13]. As a result, disease risk variants that surpassed the genome-wide significant level ($P < 5.0 \times 10^{-8}$) have been identified in a total of 26 loci outside the HLA region (Supplementary Data 1). Among these, SNPs in STAT4

✉e-mail: chikashi.terao@riken.jp

have been well established for the trans-ethnic disease association. *STAT4* has also been implicated in the association with multiple AIDs including systemic lupus erythematosus (SLE) and Rheumatoid arthritis (RA)[14]. On the other hand, the rest of the loci were more pronounced mainly in Europeans.

While the previous studies had greatly contributed to a better understanding of genetic backgrounds of SSc pathology as well as shared and population-dependent genetic architecture, the study subjects enrolled were mostly limited to European descendants; there have been only three East Asian GWASs with limited sample sizes[11,13,15]. Among these East Asian studies, the *STAT4* region was convincingly associated in two studies, while none of the loci outside the HLA region were reproducibly associated. Furthermore, it has been difficult to identify disease-associated variants with low allele frequencies in Asians because of the limited sample size and a lack of imputation reference panels containing a sufficient number of whole-genome sequencing (WGS) data from East Asian populations. Since rare susceptibility variants tend to have larger effect sizes compared to common SNPs, and importantly, may have population-specific effects, we can analyze rare variants by enrolling enough subjects including both cases and controls in the Asian population. As a result, the underlying genetic architecture of East Asian SSc, in terms of both similarity to and difference from European SSc, has not been well uncovered yet.

Here we conducted GWAS for Japanese SSc comprising a total of 114,027 subjects, consisting of 1428 cases and 112,599 controls, taking advantage of an imputation reference panel containing more than 3000 Japanese WGS data. These numbers of cases resulted in the largest Asian GWAS for SSc ever. We further conducted a trans-ethnic GWAS meta-analysis by combining our GWAS dataset with a comprehensive European meta-GWAS result and downstream analyses to clarify the genetic basis of SSc.

## Results

### Three novel risk SNPs identified in the non-HLA region by Japanese GWAS

The current study included two independent datasets, which resulted in a total of 114,108 samples (Supplementary Fig. 1). Set 1 consisted of 694 cases enrolled from the previous study[11] and 2095 controls. Set 2 consisted of newly enrolled 734 cases and 110,504 controls enrolled from the BioBank Japan (BBJ) project[16,17]. Genotype phasing and imputation were conducted after rigorous quality controls (QCs) of samples and variants (Supplementary Fig. 2). Then, an association analysis was conducted for each set, which identified one and six significant loci in Set 1 and Set 2, respectively (Supplementary Fig. 3, 4, Supplementary Data 2, 3).

Next, we combined the two datasets to maximize the statistical power of association analysis. Consequently, a total of five significant loci were identified outside the HLA region (Fig. 1a, Table 1) and three of them were novel; rs6697139 in *FCGR/FCRL* region, rs5029949 in the *TNFAIP3* region, and rs2819422 in the *AHNAK2-PLD4* region (Fig. 1b–d). All five loci showed strong associations both in Set 1 and 2 with the same effect directions (Table 1). The genomic inflation factor $\lambda_{GC}$ was 1.044 (Supplementary Fig. 5) and the intercept of linkage disequilibrium score regression (LDSC) was 1.023 showing no obvious confounding bias. The effect directions of known risk loci were shared for all the loci between previous GWASs and the present study (Supplementary Data 1), indicating a strong shared genetic architecture between European and Japanese populations in addition to the validity of the present GWAS.

We also ran the Firth regression[18] and a regression based on a generalized linear mixed model with the saddle point approximation using SAIGE[19] to address possible overestimation of the low frequent variants (such as the *FCGR/FCRL* SNP) due to the imbalance of case-control numbers, which revealed comparable results with those

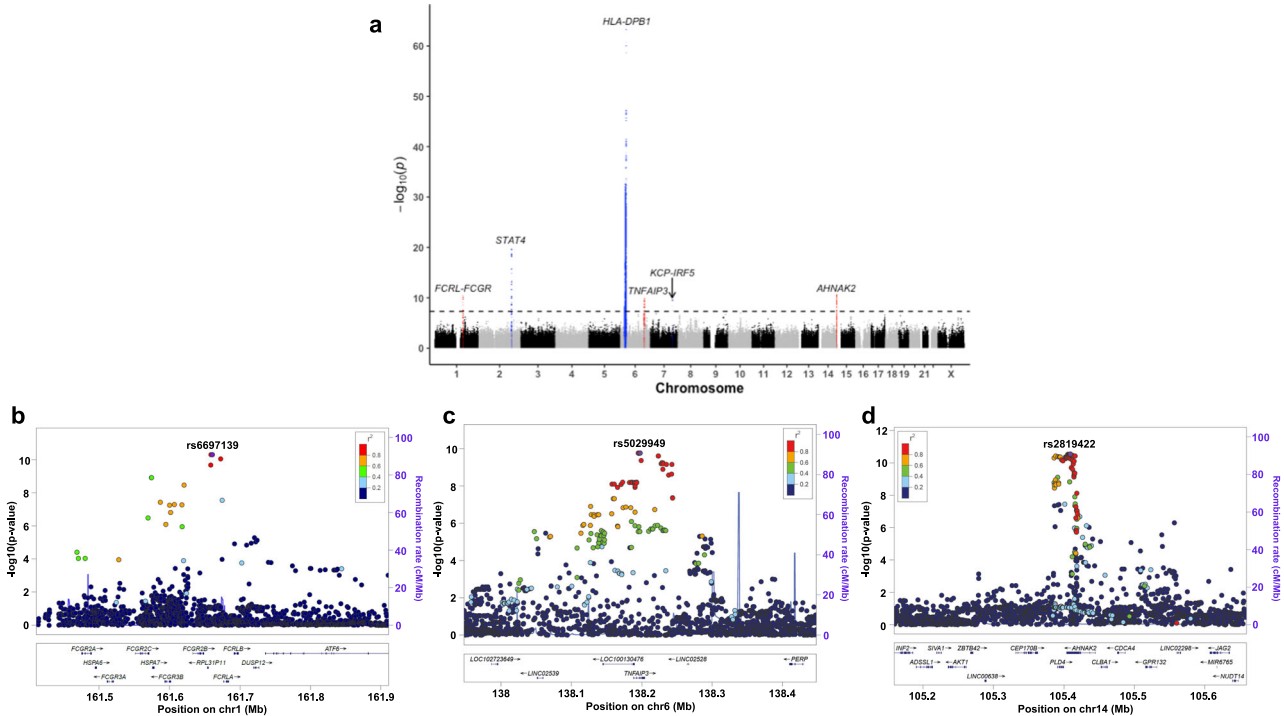

**Fig. 1 | Five significant non-HLA loci identified by the ever-largest Asian GWAS for systemic sclerosis. a** A Manhattan plot of the GWAS for systemic sclerosis (SSc) in the Japanese population. The novel risk loci are red-highlighted and SNPs with genome wide-significance (threshold $p = 5.0 \times 10^{-8}$; the black dotted line) are indicated by light green. **b–d** Regional locus zoom plots for novel single nucleotide polymorphisms (SNPs) identified by the GWAS for Japanese SSc. Each dot is colored by $r^2$ of linkage disequilibrium (LD) with the purple-colored lead SNPs indicated with texts (chromosome position). 1428 cases and 112,599 controls were analyzed by logistic regression. Source data are provided as a Source Data file.

**Table 1 | Genome-wide significant SNPs identified in GWAS of Japanese SSc**

| CHR | POS | ID | TYPE | GENE | EA | NEA | Combined dataset (Set 1 and Set 2) | | | | Set 1 | | | | Set 2 | | | | EUR | | | |
| | | | | | | | EAF (case control) | BETA | SE | P-value | EAF | BETA | SE | P-value | EAF | BETA | SE | P-value | EAF | BETA | SE | P-value |
|---|---|---|---|---|---|---|---|---|---|---|---|---|---|---|---|---|---|---|---|---|---|---|
| 1 | 161660696 | rs6697139 | intergenic | *RPL31P11-FCRLA* | T | G | 0.038 | 0.719 | 0.109 | 4.93E-11 | | 0.727 | 0.208 | 4.85.E-04 | | 0.74 | 0.150 | 7.90E-07 | 0.225 | 0.059 | 0.037 | 0.1107 |
| 2 | 191943742 | rs11889341 | intronic | *STAT4* | T | C | 0.373 | 0.37 | 0.040 | 2.51E-20 | | 0.432 | 0.069 | 3.62E-10 | | 0.33 | 0.056 | 7.57E-09 | 0.229 | 0.292 | 0.094 | 0.0018 |
| 6 | 138197506 | rs5029949 | intronic | *TNFAIP3* | G | A | 0.102 | 0.402 | 0.063 | 1.66E-10 | | 0.422 | 0.114 | 2.03E-04 | | 0.423 | 0.087 | 1.12E-06 | 0.022 | 0.291 | 0.085 | 0.0007 |
| 7 | 128575797 | rs1450734198 | intergenic | *KCP-IRF5* | T | TCTTAG CTATTGCTC | 0.163 | 0.346 | 0.055 | 2.74E-10 | | 0.206 | 0.097 | 0.0346 | | 0.44 | 0.074 | 2.40E-09 | NA | NA | NA | NA |
| 14 | 105408955 | rs2819422 | exonic[a] | *AHNAK2* | G | A | 0.312 | -0.276 | 0.041 | 2.73E-11 | | -0.33 | 0.069 | 1.55E-06 | | -0.27 | 0.058 | 2.84E-06 | 0.466 | 0.092 | 0.035 | 0.0092 |

1428 cases and 112,599 controls were analyzed by logistic regression.
*EUR* European, *CHR* chromosome, *POS* genomic position in GRCh37 coordinate, *EA* effect allele, *NEA* non-effect allele, *EAF* effect allele frequency, *BETA* beta coefficient of logistic regression, *SE* standard error.
[a]Nonsynonymous SNV AHNAK2:NM_001350929:exon7:c.T12533C:p.V4278A, AHNAK2:NM_138420:exon7:c.T12833C:p.V4278A; gene symbols are indicated in italics; the novel disease-associated SNPs are highlighted with bold

obtained by a logistic regression (Supplementary Fig. 6). The associations of genetic variations in *TNFAIP3*[20–22] and *PLD4*[22] have been described previously, but only by CGAs, while both genes have been well documented for the associations with SLE[23,24]. Thus, this is the first GWAS that identified the associations of these loci with SSc at the GWAS significance. *STAT4* and *IRF5* are associated with multiple AIDs including SSc[14].

No further significant SNPs were identified by the stepwise conditional analyses with a relaxed threshold level ($P = 1.0 \times 10^{-6}$).

## Exploration of causal SNPs and potential impacts of GWAS SNPs on the SSc pathology

Having identified significantly associated SNPs prompted us to search for potential causal variants in each locus.

We searched for potential deleterious exonic or loss-of-function (LoF) SNPs among the lead SNPs and their linked SNPs ($r^2 \geq 0.8$) and found that none of the SNPs tested were predicted to be LoF SNPs or potential deleterious exonic SNPs (Supplementary Data 4–7).

Next, we conducted fine-mapping for each significant locus and identified the plausible causal SNPs in the *FCGR/FCRL*, *STAT4*, and *IRF5* regions, where the top two variants had posterior probabilities (PPs) of more than 0.3 (Fig. 2a–e, Supplementary Data 8).

Then, we conducted a tissue-wide association study (TWAS) using the gene expression profiles in multiple tissues in the GTEx V7 (https://www.gtexportal.org/home/downloads/adult-gtex#qtl)[25] and those in six subsets of white blood cells (WBCs) (https://humandbs.biosciencedbc.jp/en/hum0099-v1)[26] to evaluate associations between SSc susceptibility and gene expression profiles of multiple tissues or cell types. A total of 26 gene-tissue/cell-type pairs were found to be significantly associated with SSc susceptibility (Supplementary Data 9). Among these, *IRF5* is a well-established risk gene for SSc. Notably, the changes of *IRF5* expression were mainly observed in various non-lymphoid tissues or organs, most of which can be involved in the disease course of fibrosis (Supplementary Data 9). Among the genes located in close proximities on chromosome 14, *AHNAK2*, *C14orf180*, *GPR132*, and *PLD4* (Fig. 2), the strongest association was observed for an increased *PLD4* expression in the spleen (TWAS $P = 6.3 \times 10^{-14}$, Supplementary Data 9). Thus, together with the previous findings of auto-immune phenotypes in *pld4* mutant mice[22], the altered expression of *PLD4* expression in immune-related cells or tissues including the spleen might confer SSc susceptibility.

We also conducted pathway analyses to measure pathological impacts by the GWAS significant SNPs. As expected, significant enrichment and a high proportion of overlapping annotated genes were observed in the SLE-related pathway followed by various immune-related pathways, such as the butyrophilin-related pathway, IL-20 family signaling, Fc γ receptor-mediated phagocytosis, adaptive immune system, JAK-STAT signaling pathway, or IFN-γ pathway (Supplementary Data 10).

## Trans-ethnic meta-analysis of Japanese and European SSc GWAS

In search of risk SNPs of SSc shared between Japanese and European populations, we estimated the correlation of the effect sizes of causal SNPs between the two populations (Supplementary Data 11). The recently published European GWAS meta-analysis summary statistics (https://www.ebi.ac.uk/gwas/publications/31672989)[12] was used as a representative European dataset. We found the trans-ethnic genetic correlation estimate (ρge) of 0.738 ± 0.418 (standard error of mean), suggesting highly shared genetic architecture between the two populations.

Then, we conducted a trans-ethnic meta-analysis by an inverse variance method with a fixed-effect model. We took a total of 3,686,421 SNPs shared between the two data sets for the meta-analysis. Although the genomic inflation factors ($\lambda_{GC}$) were 1.110 and 1.098 with and without SNPs in the HLA region, respectively (Supplementary

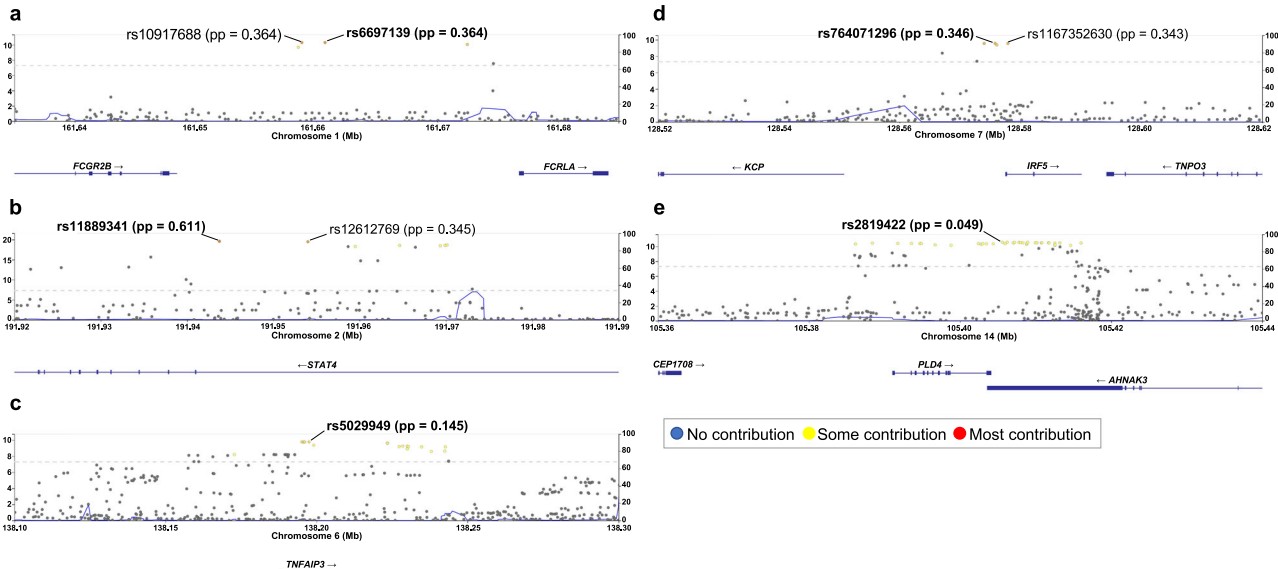

**Fig. 2 | Regional plots of credible sets fine-mapped in each GWAS locus.** The regional plot of each significant locus of the Japanese GWAS for SSc is presented. Each dot represents each SNP colored by the posterior probability (pp). Lead SNPs highlighted in bold and SNPs with pp > 0.3 are specified in each graph. **a** *FCGR/ FCRL*, (**b**) *STAT4*, (**c**) *TNFAIP3*, (**d**) *IRF5*, (**e**) *AHNAK2-PLD4*. The left *y*-axis indicates -log10 (*p*-value) and the right *y*-axis indicates recombination rate (cM/Mb). Approximate Bayesian factors calculated for each locus and 95% credible set was created based on posterior inclusion probabilities (PIPs). 1428 cases and 112,599 controls were included. Source data are provided as a Source Data file.

Fig. 7), the LDSC intercept was 1.05, indicating that the genomic inflation observed in the meta-analysis was mainly due to polygenic nature of SSc.

A total of 24 lead SNPs were identified (Fig. 3a, Table 2) and three of them, rs398390 in *LINC01980-CMC1-EOMES* region, rs10484921 in the *ESR1* region, and rs2074 in the *SLC12A5* region, were found to be novel (Fig. 3b–d). Among these novel risk SNPs, rs10484921 and rs2074 are eQTL and pQTL for *EOMES* and CD40 expression, respectively[27]. We further conducted conditional analyses by conditioning on the lead SNPs in each population followed by a meta-analysis to obtain additional independently associated SNPs, which was repeated until no further SNPs remained significant at a relaxed significant threshold ($P = 1.0 \times 10^{-6}$). We identified five more secondary and one more tertiary independently associated SNPs (Table 2). Among the lead SNPs identified in Japanese GWAS of the Japanese population that were also present in GWAS of European populations showed the same direction of effects in both populations (Supplementary Data 12). Taken together, a total of 30 signals including three novel associations in 24 regions were identified in the trans-ethnic meta-GWAS analysis of Japanese and European SSc.

Subsequently, a fine-mapping analysis was conducted to identify potential causal SNPs shared between the two populations. We found that a single SNP had the PPs close to 1.0 in eight regions (Supplementary Data 13). Furthermore, each credible set had only two SNPs in another four regions with the PPs > 0.6 for top SNPs. Among these 12 regions, a total of 5 regions, namely, *NF-κB, PRDM1-ATG5, IRF5, TNFSF4*, and *IRF8*, were those which were not successfully fine-mapped using the European meta-GWAS dataset only[12] (Supplementary Data 14), showing that the present trans-ethnic meta-analysis not only identified novel susceptibility SNPs but also successfully fine-mapped candidate causal SNPs by taking advantage of difference in LD structures between the populations. The fine-mapped variants with high PPs tended to present high prediction scores of Regulome DB (https://regulomedb.org/regulome-search/)[28]. Furthermore, according to HaploReg (https://pubs. broadinstitute.org/mammals/haploreg/haploreg.php)[29], those variants were marked by active histone marks, especially enhancer-related histone marks, H3K4me1 or H3K27ac, in blood immune cells, as well as skin fibroblasts or even in fetal lung fibroblast cell line, IMR90 (*TNFAIP3, DGKQ, TNIP1, BLK, DDX6, CSK*). Together these results demonstrate that the fine-mapping followed by the annotation of regulatory elements successfully picked up candidate causal SNPs, which can be prioritized for further functional studies.

## Transcriptional regulation of FcγR family genes via IRF8 binding to rs10917688-containing cCRE

For further detailed downstream analyses, we focused on the *FCGR/ FCRL* region since this region showed a penetrating association with SSc, especially in the Asian population, one of the unexplored populations and a main data source of findings in the current study. rs10917688, one in complete LD with the lead SNP in the *FCGR/FCRL* region (Fig. 1a, b, Table 1) with the high effect size (OR 2.05), was found to be positioned within a candidate cis-regulatory element (cCRE) nearby a cluster of Fcγ receptor (FcγR) family genes according to the ENCODE database (https://www.encodeproject. org)[30,31]. This is in line with most GWAS signals found in intronic or intergenic regions, in which regulatory elements for gene expression are frequently located. Among such SNPs, a transcription factor (TF)-binding motif analysis for the cCRE containing rs10917688 identified a total of 28 TF binding motifs in the cCRE. We noted that more than half of these TFs were known for immune cell differentiation or function, and they tended to be ranked higher than non-immune-related TFs (Supplementary Data 15). Among these immune-related TFs, IRF8 was the only TF predicted to bind to the sequence spanning rs10917688 with statistical significance (Fig. 4a and Supplementary Data 15), with higher binding probability on the risk allele, indicating that altered IRF8 binding can affect SSc susceptibility. Notably, one of the lead SNPs identified in the trans-ethnic meta-GWAS, rs11117420, is located in the *IRF8* region (Table 2). Furthermore, the association of rs10917688 was observed only in the presence of the risk genotype (GG) of rs11117420 (Fig. 4b), indicating an IRF8-dependent association of rs10917688. IRF-targeted gene enrichment, which included *FCGR2B* and *FCGR2C*, was also identified by gene set analysis (Supplementary Data 16), further supporting the association between IRF8 and rs10917688.

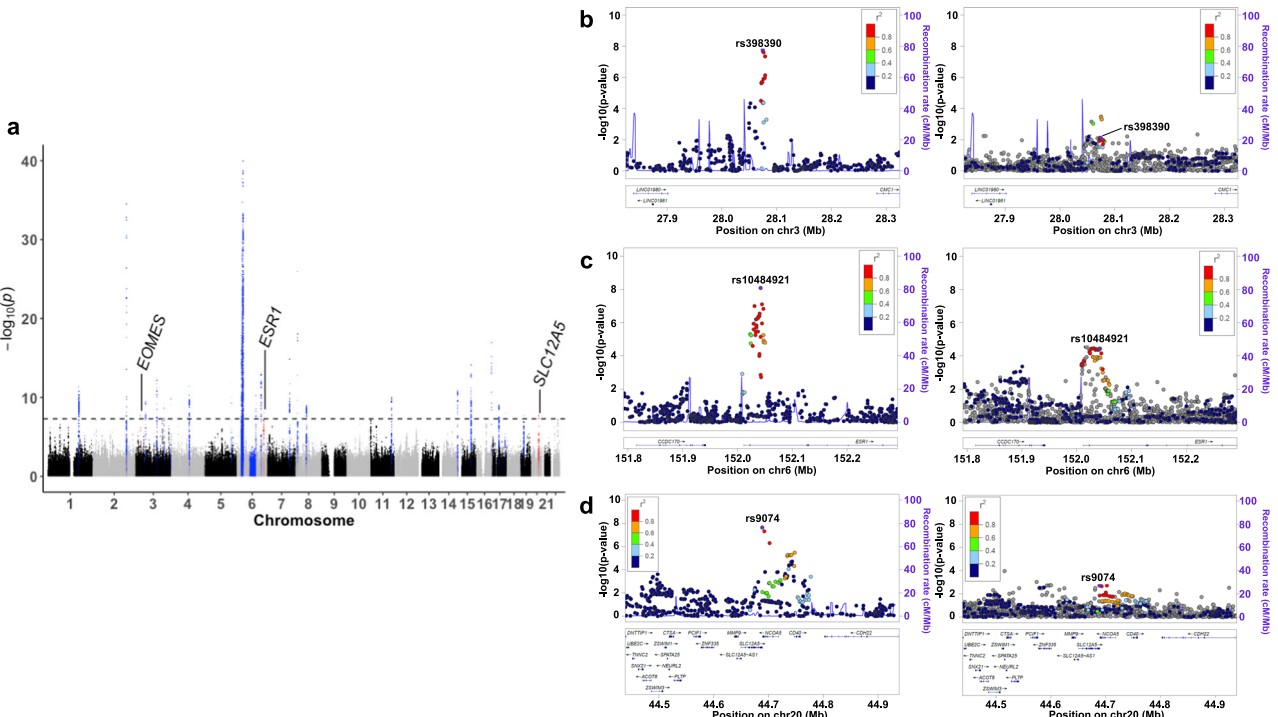

**Fig. 3 | Twenty-eight significant signals including four novel SNPs identified by the meta-analysis for the Japanese and the European GWASs. a** A Manhattan plot representing the lead signals identified in the meta-analysis for GWASs of European and Japanese SSc. **b–d** Regional locus zoom plots for novel significant SNPs identified by the trans-ethnic meta-GWAS (A). For each SNP, regional plots for European and Japanese populations are presented on the left and right, respectively. Inverse-variance fixed effect model was utilized for a meta-analysis of Japanese (1428 cases and 112,599 controls) and European (9095 cases and 17,584 controls) GWAS datasets. Source data are provided as a Source Data file.

Then, we explored the cell-type specificity of the observed potential IRF8 binding to the cCRE. We identified enhancer-like histone marks, H3K27ac in primary CD8+ memory T cells and H3K4me1 in primary B cells at this locus (Supplementary Data 17). We further examined if rs10917688 is an expression quantitative trait locus (eQTL) for neighboring genes (Fig. 1b) using the Japanese eQTL dataset of six WBC subpopulations[26]. For *FCGR2A* and *FCGR2B*, of which expressions were found in B cells, Monocytes, and NK cells in the dataset, the effect allele (T) of rs10917688 showed a trend of less *FCGR2A* and *FCGR2B* expressions (Supplementary Fig. 8a, b). The same trend was observed only in CD8+ T cells for *FCGR3A* and *FCRLB* expressions (Supplementary Fig. 8c, d). Of note, referring to the ImmuNexUT database consisting of the eQTL dataset of 28 immune cell types obtained from patients with 10 immune-mediated diseases and healthy subjects[32], we found that *FCGR2B* expression was pronounced in B cells, among which unswitched memory (USM) B cells expressed the highest in healthy subjects. Furthermore, USM B cells from Japanese SSc patients expressed less *FCGR2B* than those of healthy subjects. This trend was consistent in other B cell subsets including switched memory B cells, double-negative B cells, and plasmablasts. Furthermore, one SNP, rs10917698, which is in a perfect LD with rs10917688, was a significant eQTL in plasmablasts ($p = 5.97 \times 10^{-7}$, β = −0.571). On the other hand, GTEx database (v8.1), which is based on samples from European descendants and hence different LD structure from that of the Japanese population, only revealed that rs10917688 was eQTL for *FCRLB* expression in whole blood (normalized effect size (NES) −0.20, $p = 3.1 \times 10^{-6}$). Intriguingly, both rs6697139 and rs10917688 were pQTL for FCGR2A and 2B expression in blood plasma although the origin of cell type was unclear[33].

Together these results indicate that rs10917688 may affect the expression of nearby FcγR family gene(s) by altering the binding affinity of IRF8 in the target cells, especially B cells.

## Different genetic architectures among the major clinical and serological subtypes

Next, we explored genetic associations in major clinical/serological subtypes of SSc, lcSSc, dcSSc, ACA-positive SSc, ATA-positive SSc, SSc complicated with ILD (ILD-SSc) (Supplementary Data 18). We found that there was a large difference in association patterns between the diffuse forms of SSc (dcSSc and ATA-positive SSc) and the limited forms of SSc (lcSSc and ACA-positive SSc); the solely strong association of the *HLA* locus was identified in the diffuse forms of SSc, while the associations of *HLA* was much weaker and multiple loci outside the *HLA* region including *STAT4* locus were associated with the limited forms of SSc (Fig. 5a, Supplementary Fig. 9, Supplementary Data 19). Effect sizes of the non-HLA lead SNPs in each subset or whole SSc were comparable among different subsets except for those of three SNPs, rs5862323, rs11177005, and rs79834248, which were associated with the ACA-positive SSc; the effect sizes of these SNPs were comparable only with those of lcSSc (Fig. 5b). Likewise, the association of the lead SNP in the *FCGR/FCRL* locus, rs6697139, was significant only in whole SSc and the effect sizes were almost identical except for ACA-positive SSc (Supplementary Fig. 10, Supplementary Data 20). We also observed the same association pattern in ILD-SSc with diffuse forms or ATA-positive SSc reflecting a higher incidence of ILD in these subtypes. Though marginally significant, rs57919238 in the *NBEA* locus was identified only in ILD-SSc. Finally, as indicated in the previous Caucasian study[20], the association of *TNFAIP3* locus was much stronger in ILD-SSc with the higher effect size and the smaller *p*-value than SSc without ILD in comparison with control subjects. Though not significant due to the limited sample size, ILD-SSc had a higher effect size than SSc without ILD in the intra-case comparison (Supplementary Data S21).

Together these results highlight the different genetic architectures between the diffuse and the limited forms of SSc.

**Table 2 | Thirty SNPs associated with SSc in a meta-analysis comparing signals in European and Japanese GWASs**

| CHR | POS | ID | Type | Gene | EA | NEA | Meta-analysis | | | | | JPN | | | | EUR | | | |
|---|---|---|---|---|---|---|---|---|---|---|---|---|---|---|---|---|---|---|---|
| | | | | | | | BETA | SE | P-value | Q | I | EAF (case control) | BETA | SE | P-value | EAF | BETA | SE | P-value |
| **Primary** | | | | | | | | | | | | | | | | | | | |
| 1 | 167420425 | rs2045526 | intronic | CD247 | G | T | -0.076 | 0.028 | 2.22E-10 | 0.012 | 84.1 | 0.125 | -0.021 | 0.057 | 0.398 | 0.390 | -0.092 | 0.032 | 1.31E-11 |
| 1 | 173243581 | rs844663 | ncRNA_intronic | TNFSF4/LOC100506023 | C | T | 0.076 | 0.025 | 4.96E-12 | 0.870 | 0 | 0.281 | 0.079 | 0.043 | 2.13E-05 | 0.276 | 0.075 | 0.032 | 5.05E-08 |
| 2 | 191959489 | rs4853458 | intronic | STAT4 | A | G | 0.140 | 0.026 | 1.91E-35 | 0.310 | 3.11 | 0.425 | 0.153 | 0.039 | 5.01E-19 | 0.231 | 0.129 | 0.035 | 4.86E-18 |
| 3 | 28074673 | **rs398390** | **intergenic** | **LINCO1980-CMC1-EOMES** | **C** | **T** | **0.048** | **0.020** | **1.66E-08** | **0.846** | **0** | **0.450** | **0.045** | **0.039** | **0.0076** | **0.487** | **0.049** | **0.023** | **6.34E-07** |
| 3 | 58131515 | rs7355798 | intronic | FLNB | T | C | 0.061 | 0.022 | 3.40E-10 | 0.377 | 0 | 0.076 | 0.088 | 0.073 | 0.0054 | 0.243 | 0.058 | 0.024 | 1.24E-08 |
| 3 | 119116150 | rs9884090 | intronic | ARHGAP31 | A | G | -0.074 | 0.024 | 6.15E-13 | 0.469 | 0 | 0.281 | -0.062 | 0.042 | 6.96.E-04 | 0.155 | -0.078 | 0.029 | 1.89E-10 |
| 3 | 159733527 | rs589446 | ncRNA_intronic | IL12A-AS1 | T | G | -0.060 | 0.023 | 8.02E-10 | 0.087 | 65.8 | 0.079 | -0.011 | 0.070 | 0.724 | 0.350 | -0.066 | 0.024 | 1.95E-10 |
| 4 | 965779 | rs11724804 | intronic | DGKQ | A | G | 0.057 | 0.021 | 1.86E-10 | 0.047 | 74.6 | 0.438 | 0.025 | 0.043 | 0.185 | 0.441 | 0.067 | 0.024 | 5.31E-11 |
| 4 | 103449041 | rs230534 | intronic | NFKB1 | T | C | 0.063 | 0.021 | 4.02E-12 | 0.775 | 0 | 0.398 | 0.067 | 0.041 | 1.78.E-04 | 0.338 | 0.061 | 0.025 | 5.38E-09 |
| 5 | 150455732 | rs3792783 | intronic | TNIP1 | G | A | 0.069 | 0.023 | 2.15E-12 | 0.052 | 73.4 | 0.252 | 0.038 | 0.043 | 0.0439 | 0.158 | 0.081 | 0.027 | 2.42E-12 |
| 6 | 106568034 | rs548234 | intergenic | PRDM1-ATG5 | C | T | 0.071 | 0.025 | 8.06E-11 | 0.198 | 39.7 | 0.378 | 0.088 | 0.039 | 2.17E-07 | 0.314 | 0.059 | 0.034 | 2.86E-05 |
| 6 | 138195151 | rs5029937 | intronic | TNFAIP3 | T | G | 0.159 | 0.050 | 2.15E-13 | 0.401 | 0 | 0.102 | 0.175 | 0.063 | 1.67E-10 | 0.024 | 0.137 | 0.086 | 1.17E-04 |
| 6 | 152042260 | **rs10484921** | **intronic** | **ESR1** | **A** | **C** | **0.060** | **0.024** | **8.30E-09** | **0.083** | **66.6** | **0.150** | **0.095** | **0.053** | **3.58E-05** | **0.243** | **0.051** | **0.028** | **1.20E-05** |
| 7 | 128573967 | rs4728142 | intergenic | KCP-IRF5 | A | G | 0.092 | 0.027 | 1.61E-15 | 0.006 | 86.6 | 0.142 | 0.148 | 0.054 | 2.83E-10 | 0.449 | 0.074 | 0.031 | 2.24E-08 |
| 8 | 11343973 | rs2736340 | intergenic | FAM167A-BLK | C | T | -0.095 | 0.020 | 1.06E-26 | 0.832 | 0 | 0.285 | -0.099 | 0.046 | 6.23E-07 | 0.244 | 0.094 | 0.023 | 3.33E-21 |
| 8 | 61396829 | rs2013112 | intergenic | LINCO1301-RAB2A | T | C | -0.049 | 0.018 | 1.05E-09 | 0.611 | 0 | 0.385 | -0.056 | 0.039 | 9.78E-04 | 0.379 | -0.046 | 0.021 | 2.66E-07 |
| 11 | 118642085 | rs10892286 | intronic | DDX6 | C | A | -0.066 | 0.024 | 1.31E-10 | 0.053 | 73.3 | 0.115 | -0.020 | 0.060 | 0.447 | 0.201 | -0.075 | 0.026 | 2.12E-11 |
| 14 | 105407208 | rs11851053 | exonic | AHNAK2* | C | T | -0.053 | 0.032 | 1.70E-11 | 0.000 | 94.2 | 0.322 | -0.116 | 0.042 | 2.81E-11 | 0.466 | -0.037 | 0.056 | 2.27E-05 |
| 15 | 75077367 | rs1378942 | intronic | CSK | A | C | -0.066 | 0.020 | 9.15E-15 | 0.159 | 49.6 | 0.186 | -0.039 | 0.048 | 0.0626 | 0.387 | -0.071 | 0.022 | 1.84E-14 |
| 16 | 85971922 | rs11117420 | intergenic | IRF8-LINCO1082 | C | G | -0.094 | 0.025 | 1.42E-17 | 0.970 | 0 | 0.096 | -0.095 | 0.065 | 6.97.E-04 | 0.195 | -0.094 | 0.028 | 3.82E-15 |
| 17 | 38063381 | rs883770 | intronic | GSDMB | T | C | 0.048 | 0.018 | 1.17E-09 | 0.412 | 0 | 0.278 | 0.035 | 0.042 | 0.0564 | 0.498 | 0.051 | 0.021 | 4.79E-09 |
| 17 | 73224639 | rs1005714 | intronic | NUP85 | G | C | -0.065 | 0.026 | 9.59E-09 | 0.399 | 0 | 0.151 | -0.042 | 0.069 | 0.166 | 0.200 | -0.069 | 0.029 | 1.87E-08 |
| 19 | 18193191 | rs2305743 | intronic | IL12RB1 | A | G | -0.062 | 0.025 | 2.37E-08 | 0.005 | 87.5 | 0.213 | -0.012 | 0.048 | 0.579 | 0.199 | -0.081 | 0.031 | 4.64E-10 |
| 20 | 44688665 | **rs9074** | **UTR3** | **SLC12A5\*\*** | **A** | **G** | **0.050** | **0.021** | **2.21E-08** | **0.826** | **0** | **0.344** | **0.054** | **0.040** | **0.0018** | **0.260** | **0.049** | **0.025** | **3.35E-06** |
| **Secondary** | | | | | | | | | | | | | | | | | | | |
| 1 | 173318664 | rs10912594 | ncRNA_intronic | TNFSF4/LOC100506023 | C | G | -0.062 | 0.021 | 6.41E-12 | 0.980 | 0 | 0.309 | -0.062 | 0.050 | 0.002085 | 0.3439 | -0.062 | 0.023 | 8.13E-10 |
| 2 | 191940451 | rs7601754 | intronic | STAT4 | A | G | -0.072 | 0.027 | 1.14E-09 | 0.155 | 50.6 | 0.136 | -0.102 | 0.058 | 2.03E-05 | 0.1809 | -0.062 | 0.031 | 4.75E-06 |
| 3 | 58375286 | rs4076852 | intronic | PXK | A | G | 0.051 | 0.021 | 1.25E-08 | 0.036 | 77.3 | 0.262 | 0.014 | 0.049 | 0.463 | 0.2604 | 0.061 | 0.023 | 1.71E-09 |
| 6 | 106734040 | rs633724 | intronic | ATG5 | T | C | 0.044 | 0.018 | 2.48E-08 | 0.257 | 22.1 | 0.504 | 0.028 | 0.040 | 0.0858 | 0.3549 | 0.049 | 0.021 | 5.87E-08 |
| 7 | 128722514 | rs17340646 | intergenic | TPI1P2-LOC407835 | G | T | 0.080 | 0.033 | 1.94E-08 | 0.830 | 0 | 0.0181 | 0.097 | 0.204 | 0.241 | 0.2078 | 0.079 | 0.033 | 3.84E-08 |
| **Tertiary** | | | | | | | | | | | | | | | | | | | |
| 2 | 191534372 | rs16832798 | intronic | NAB1 | C | T | 0.050 | 0.023 | 5.15E-07 | 0.2394 | 27.8 | 0.499 | 0.034 | 0.040 | 0.0377 | 0.1481 | 0.059 | 0.029 | 2.37E-06 |

Inverse-variance fixed effect model was utilized for a meta-analysis of Japanese (1428 cases and 112,599 controls) and European (9095 cases and 17,584 controls) GWAS datasets.

JPN Japanese, EUR European, CHR chromosome, POS genomic position in GRCH37 coordinate, EA effect allele, NEA non-effect allele, EAF effect allele frequency, BETA beta coefficient of logistic regression, SE standard error; * synonymous SNV.

AHNAK2:NM_001350929:exon7:c.A14280A:p.V4760V, AHNAK2:NM_138420:exon7:c.A14580A:p.V486OV; ** NM_001134771:c.*2421 A > G,NM_020708:c.*2421 A > G; gene symbols are indicated in italics;the novel disease-associated SNPs are hilighted with bold.

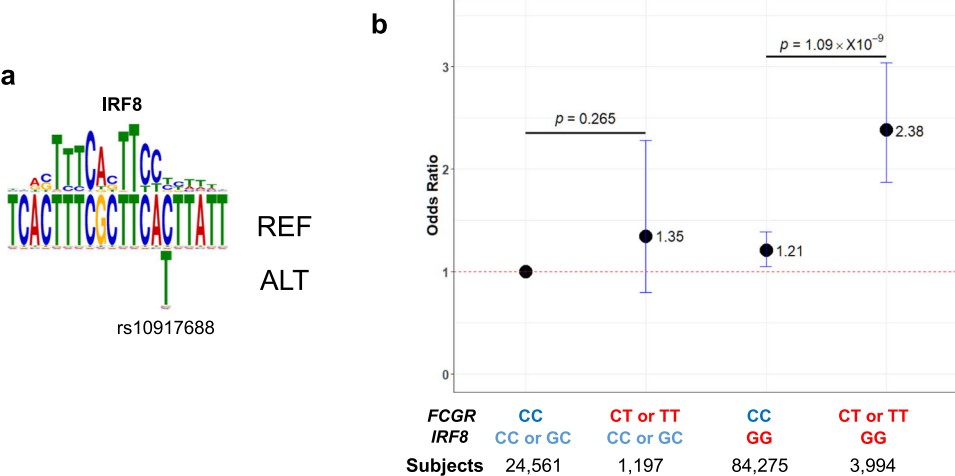

**Fig. 4 | An association of rs10917688 on IRF8 binding to the candidate cis-regulatory element. a** Relative positions of the predicted IRF8 binding site within the candidate cis-regulatory element (cCRE) is presented. The rs10917688 with the reference (REF: C) and alternative (ALT: T) alleles are indicated. **b** Odds ratio (OR) of each genotype combination relative to the control genotypes CC for rs10917688 in *FCGR/FCRL* and CC or GC for rs11117420 in *IRF8* is presented. Protective and risk genotypes are colored blue and red, respectively. Mean ORs and 95% confidence intervals are presented by black dots and blue bars, respectively. *T*-test was utilized and two-sided *p*-values were calculated. The sample number of each genotype is indicated at the bottom of the panel. Source data are provided as a Source Data file.

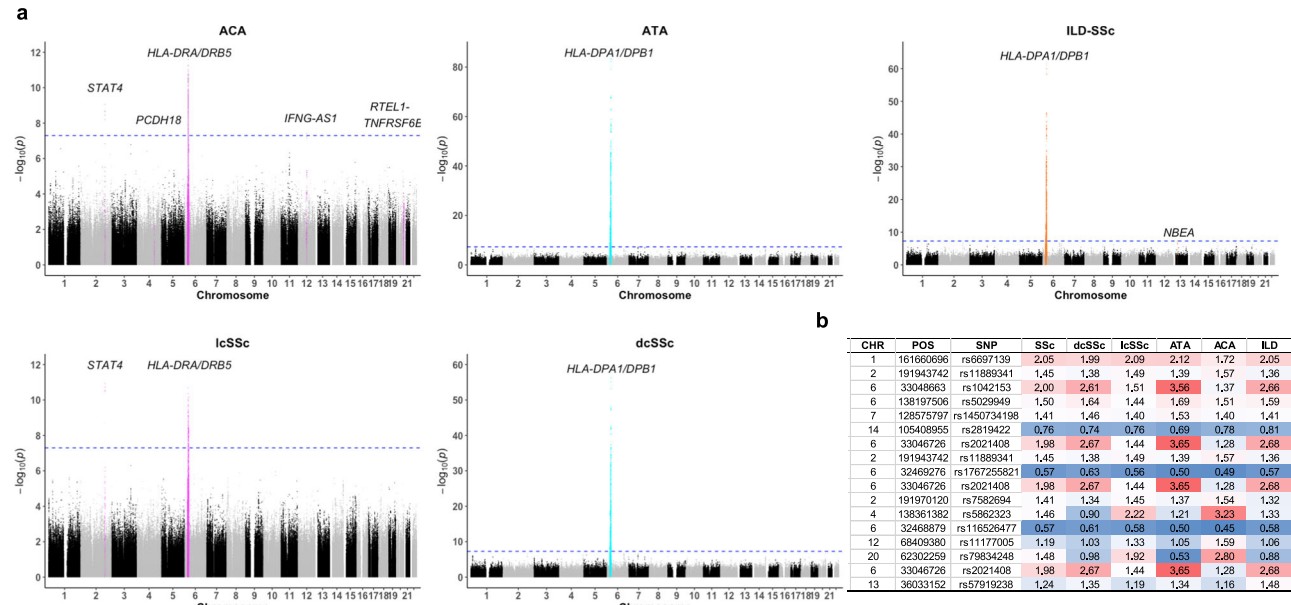

**Fig. 5 | Different patterns of association observed between the diffuse and the limited forms of SSc. a** Association test results by logistic regression for the major clinical phenotypes, lcSSc (*n* = 679), dcSSc (*n* = 575), ACA-positive SSc (*n* = 429), ATA-positive SSc (*n* = 463), and SSc complicated with ILD (*n* = 625) are presented. The blue lines indicate genome-wide significance (*p* = 5.0 × 10⁻⁸) threshold line. **b** A heatmap of the effect sizes (odds ratios) of lead variants for each subset is presented. Source data are provided as a Source Data file.

## Heritability enrichment in autoimmune-related traits/cells and polygenic features

Since it is common that genetic architectures are shared among several different diseases or traits[34], we measured the genetic correlation between SSc and SLE[35] or 47 target complex diseases of BBJ[17]. As we expected, there was a significant genetic correlation between SSc and SLE, followed by RA (without statistical significance, Supplementary Data 22). We also measured genetic correlations between SSc and 60 quantitative traits adopted in the BBJ project[16,17] and found none of the tested traits were genetically correlated with Japanese SSc (Supplementary Data 23).

Next, we conducted a partitioned heritability enrichment analysis and found that the heritability was mostly enriched in the connective tissues and bones followed by the hematopoietic tissues in the Japanese SSc, while the heritability was significantly and highly enriched in the hematopoietic tissues in the European SSc (Fig. 6a, Supplementary Data 24). The cell-type-based partitioned heritability of active histone marks, H3K9ac, H3K27ac, H3K4me3, and H3K4me1, showed that H3K4me1, one of the enhancer-related histone marks, was significantly and highly enriched in primary CD19⁺ B cells followed by CD4⁺ effector T cells in Japanese SSc (Supplementary Data 25). The same trends were also observed in European SSc with higher enrichment in CD4⁺ regulatory T cells and Th17 cells followed by B cells to a lesser extent (Fig. 6b, Supplementary Data 26).

We further applied the gchromVAR[36] to our Japanese dataset and found that B cell was the only significant cell type with enrichment of

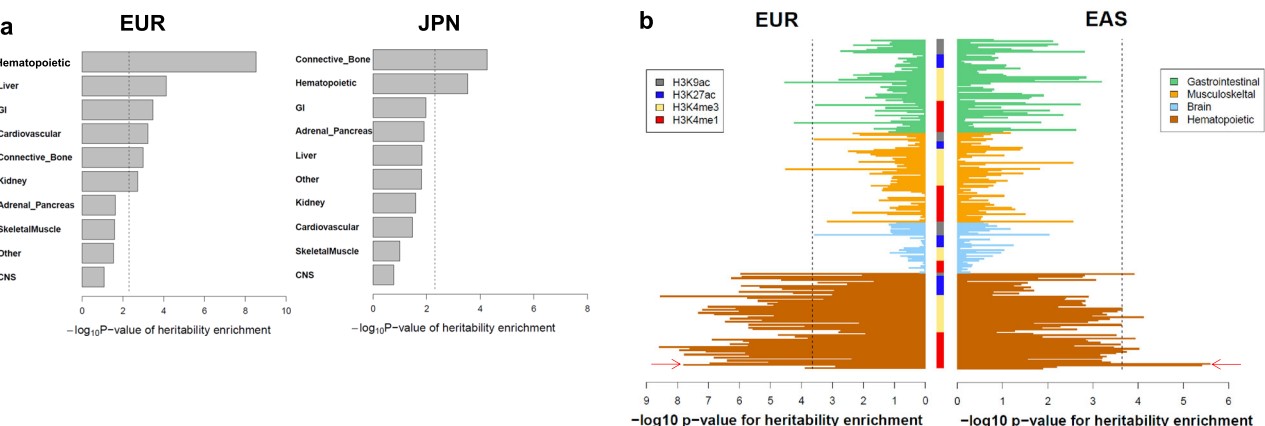

**Fig. 6 | Partitioned heritability enrichment analysis for European and Japanese SSc.** Heritability enrichment of active histone marks (H3K9ac, H3K27ac, H3K4me3, H3K4me1) in different tissue types (**a**) corresponding to Supplementary Data 20 and cell types (**b**) corresponding to Supplementary Data 21 and 22 by LD-score regression are presented. The dashed lines indicate the thresholds of significance based on the Bonferroni correction ($P < 0.05/10$ for (**a**), $P < 0.05/220$ for (**b**)). Primary B cells are indicated by red arrows. EUR European (9095 cases and 17,584 controls); JPN Japanese (1428 cases and 112,599 controls). Source data are provided as a Source Data file.

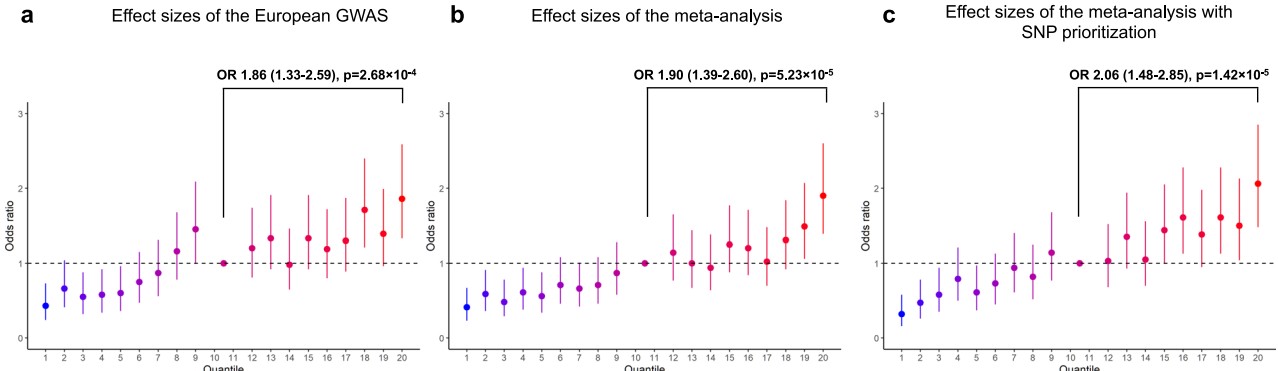

**Fig. 7 | Application of polygenic risk scores to Japanese SSc.** Set 2 Japanese samples (734 cases and 110,504 controls) are stratified into 20 quantiles based on the individual polygenic risk scores (PRSs) calculated without prioritizing SNPs with the use of effect sizes of the European GWAS (**a**), without prioritizing SNPs with the use of effect sizes of the meta-analysis of the European (9095 cases and 17,584 controls) and Set 1 Japanese (694 cases and 2095 controls) dataset (**b**), or by prioritizing the top 5% of SNPs for IRF8-binding in RAMOS cells identified by IMPACT (see Methods for detail) and the lead SNPs of the meta-analysis with the use of the effect sizes of the meta-analysis (**c**). The odds ratio, the 95% confidence intervals, and the p-value of the top 5% quantile relative to those of median quantiles (the 10th and 11th quantile) are presented. The significant threshold is determined by Bonferroni correction (0.05/18). Source data are provided as a Source Data file.

genetic variations in chromatin accessibility regions among 16 blood cells tested (Supplementary Data 27).

Together these results clearly show the polygenic architecture of SSc, which is similar to other AIDs, with strong heritability enrichment in lymphohematopoietic systems, especially in B cells.

**Prediction of SSc development by polygenic risk scores**
Having confirmed the heritability and the polygenicity of SSc shared between the European and the Japanese populations, we constructed polygenic risk scores (PRSs) using β coefficients of the European GWAS taking account of Japanese LD structures and applied them to our Japanese datasets to examine the predictive performance for the SSc development. Using the best predictive parameter set of GWAS p-value ($P_T$) and $r^2$ of LD determined by the Japanese Set 1 dataset ($r^2 = 0.4$, $P_T = 5.0 \times 10^{-6}$; Supplementary Data 28), the disease risk was tested in Set 2, resulting in the AUC of 0.593 and the Nagelkerke value of 0.0092 (OR 1.336, 95% CI 1.263–1.424, $p = 9.98 \times 10^{-19}$). The performance was much better than that constructed by β coefficients of the Set1 Japanese dataset (AUC of 0.519 and Nagelkerke value of 0.00055 in Set2 with the same parameter set as above). The subjects in the top 5%

quantile had significantly higher OR for SSc susceptibility compared to those in the median quantile (Fig. 7a). When we used the effect sizes of a meta-analysis of the European and the Japanese Set 1 datasets, the predictive performance further improved (AUC from 0.593 to 0.604 and Nagelkerke pseudo $R^2$ from 0.0092 to 0.0118; Supplementary Data 29) and the subjects in the top 5% quantile had a significantly higher risk than those in the median PRS (Fig. 7b) with the higher OR compared to one observed in the comparison with the use of the European GWAS effect sizes (Fig. 7a). These results highlight a shared genetic architecture of SSc between European and Japanese populations.

Next, we tested if the PRS predicts subsets of SSc better than a whole set of SSc using the best parameter sets determined above. Intriguingly, predictive performances were slightly higher for lcSSc or ACA-positive SSc, suggesting a more polygenic nature of the limited cutaneous types of SSc. Furthermore, in the intra-case setting, the PRS significantly discriminated SSc patients with ACA from those without ACA (Supplementary Data 30). We also examined a possible correlation between PRS and the age of SSc onset, which revealed no significant correlation (Supplementary Data 31).

Since the present findings consistently demonstrated the significant roles of B cells and IRF8 in SSc pathology, we hypothesized that the integration of functional annotations in B cells would improve the fitness of PRS. We obtained the top 5% of SNPs for IRF8 binding in one of the B cell lines, RAMOS cells, by IMPACT software[37,38] and measured an improvement in the predictive performance of PRS. As expected, prioritizing these top 5% SNPs plus the lead SNPs of the meta-analysis further improved the predictive performance of PRS (AUC from 0.604 to 0.610 and Nagelkerke pseudo $R^2$ from 0.0117 to 0.0130; Supplementary Data 32) and the top 5% quantile subjects had a significantly higher risk of SSc than those with the median PRS (Fig. 7c) with the higher OR compared to those observed without the SNP prioritization (Fig. 7a, b). Together these data further support the importance of IRF8 and B cells in SSc development as well as better trans-ancestral portability of PRS by prioritizing SNPs annotated according to TF-binding in tissue and cell-type specific manners.

## Discussion

In the present study, we conducted GWAS of Japanese SSc comprising a total of 114,027 subjects consisting of 1428 cases and 112,599 controls, resulting in the largest Asian GWAS for SSc ever, and identified three novel significant loci outside the HLA region, which needs a separate analysis due to the complex LD structure and thus was remained for an extension study in the future. As presented in Supplementary Data 1, the effect sizes of the previously identified risk variants were concordant, demonstrating the validity of the current study. Previously, candidate gene analyses (CGA) have reported multiple potential susceptibility loci to SSc. An association of the *TNFAIP3* locus has repeatedly been implicated by multi-ethnic CGAs[20–22]; however, the association exceeding the GWAS significant level has never been reported by any of the GWASs. Notably, the lead SNP, rs5029949, is in almost perfect LD with the SNP previously reported for its association in European SSc, rs5029939 (*D'* 1.0, $r^2$ 0.9148)[20]. On the other hand, the *PLD4* locus had been known as one of the risk loci for Japanese SLE and was found also to be associated with Japanese SSc, but without fulfilling the GWAS significance[22]. Both these loci were identified for their genome-wide significant associations for the first time, showing a substantial improvement in statistical power in the present study. Notably, one of the SNPs, rs6697139 located at the intergenic region of FcγR family genes as well as its complete LD SNP, rs10917688, had a strong effect size (OR ~ 2.0) and were found to be plausible causal SNPs by fine-mapping. Considering the much higher MAF but with no significant association in the European dataset, this Japanese-specific loci was worth further investigation. Notably, rs10917688 is positioned within a cCRE and likely to be a part of binding motifs of IRF8, a key TF for the developmental trajectory of B cells[39] as well as dendritic cells, macrophages, and NK cells[40]. One of the enhancer-related histone marks, H3K4me1, was identified in B cells and heritability of active histone marks was enriched in B cells both in European and Japanese populations, suggesting IRF8-*FCGR/FCRL* axis in B cells might be a pathological mechanism in SSc.

FcγRs are expressed on the surface of both innate and adaptive immune cells and confer immune modulatory responses by binding the Fc portion of immunoglobulin G (IgG). Based on their binding affinity, FcγRs are divided into low-affinity and high-affinity FcγRs and there are five low-affinity FcγRs, FcγRIIa, FcγRIIb, FcγRIIc, FcγRIIIa, and FcγRIIIb, each of which is encoded by *FCGR2A*, *FCGR2B*, *FCGR2C*, *FCGR3A*, and *FCGR3B*, respectively[41]. Due to the close proximities of FcγR gene locations and segmental duplications (SD), loci with two or more highly similar and duplicated regions, it is difficult to determine which gene is critically affected by a causal SNP for SSc development. SD loci are enriched for immune genes and often show copy number variations (CNVs)[41]. Indeed, CNVs in the *FCGR* region have been suggested to be associated with multiple AIDs including RA, SLE, celiac disease, and inflammatory bowel disease[41]. However, the association of

CNVs with SSc has never been reported, and thus the effect of the SNPs rather than CNVs is likely to be associated with SSc. In addition, the association was not significant despite higher MAFs in European populations, suggesting the association of these loci is specific to Japanese population, which may be independent of CNV.

On the other hand, our eQTL analysis using the Japanese eQTL dataset of six WBC subpopulations[26] showed a trend of decreased *FCGR2A* and *FCGR2B* expression in relevant cell types, B cells, NK cells, and Monocytes. FcγRIIa, CD32a, is expressed in monocytes, neutrophils, and eosinophils, and mediates phagocytosis of opsonized antigens or immune complexes. FcγRIIa has Immunoreceptor Tyrosine-based Activation Motifs (ITAMs) and Immunoreceptor Tyrosine-based Inhibitory Motifs (ITIMs) and thus can be functionally divergent. A well-known functionally relevant SNP, rs1801274, is a nonsynonymous mutation, which alters arginine I to histidine (H) at amino acid position 131 of the extracellular domain and confers binding to IgG2 and IgG3 with higher affinity than RR receptors[42]. The SNP has been implicated for the susceptibility to SLE, Kawasaki disease, cystic fibrosis, and several infectious diseases including invasive pneumococcal or meningococcal disease, severe malaria, dengue fever, respiratory syncytial virus, and SARS-CoV[42]. FcγRIIb, CD32b, is a solely inhibitory receptor among FcγRs expressed mainly on B cells but is also expressed on dendritic cells, macrophages, and mast cells. It inhibits phagocytosis of immune complexes and antibody production by B cells[43]. *FCGR2B* has been implicated for its association with multiple AIDs including RA[44], SLE[45,46], type I diabetes[47], and IgG4-related disease[48]. Although the genetic association of *FCGR2B* with SSc has never been reported, one small study observed higher levels of anti-FcγRIIB/C antibodies in sera of Japanese dcSSc patients compared to those of lcSSc or non-SSc controls[49]. Considering the roles in immunoregulatory functions and hence the association with various immune-related diseases, the observed significant association of *FCGR/FcRL* variation with SSc may be one of the shared genetic architectures among immune-related diseases including SLE.

Interferon-regulatory factor 8 (IRF8), also known as interferon consensus sequence binding protein (ICSBP), is a TF exclusively expressed in hematopoietic lymphoid and myeloid cells. *IRF8* deficiency in humans was previously documented and the subjects suffered from severe immunodeficiency due to depletion or impaired functions of dendritic cell subsets, monocytes, and NK cells[50]. Sequence variants near *IRF8* have repeatedly been identified as risk factors for various AIDs including SSc[12] according to the previous GWASs. Indeed, the importance of *IRF8* variations and the expression change have been highlighted by several pieces of literature. Arismendi et al investigated potential associations of sixteen PBC-susceptibility SNPs with SSc and found an association between rs11117432 and SSc susceptibility with the stronger association with lcSSc[51]. On the other hand, Ototake et al found a negative correlation between *IRF8* expression in monocytes of dcSSc subjects and the modified Rodnan skin thickness score. They also found that monocyte/macrophage-specific IRF8 knock-out mice presented accelerated fibrotic phenotypes[52]. IRF8 has also been implicated in early B-cell development, Igk rearrangement, germinal center formation, and plasma cell generation[53]. Several mouse experiments showed that IRF8 worked together with other TFs, such as PU.1, IRF4, IKAROS, or E2A, to modulate lineage specification, commitment, and differentiation in B cells[54,55]. An intronic variant, rs8057456, was one of GWAS significant variants for serum immunoglobulin levels in the GWAS of Scandinavian populations[56]. Together these studies strongly suggest that genetic variations of *IRF8* in B cells can modulate susceptibility to autoimmunity including SSc. The present study indicated that IRF8 may bind to a motif containing *FCGR/FCRL* variant, rs10917688, within a cCRE, which has never been reported so far. Furthermore, our genotype combination analysis (Fig. 4b) revealed that IRF8-binding to this motif is indispensable for the risk effect of rs10917688, supporting an

interactive effect of IRF8 and the *FCGR/FCRL* variant. Although we were not able to identify eGene(s) for rs10917688 and thus the result has not been fully convinced, further independent studies for both East Asians and other populations and validation experiments such as reporter assays or single nucleotide editing approaches will validate our findings as well as clarify more precise molecular mechanisms.

As can be observed in other AIDs, our study revealed that SSc has also polygenic architecture. Intriguingly lcSSc or ACA-positive SSc tended to more fit the PRS than diffuse forms of SSc, implying different genetic backgrounds and hence pathological mechanisms between these distinctive phenotypes. Indeed, we observed the stronger association of HLA genes in diffuse forms of SSc, while limited forms of SSc showed significant associations with multiple SNPs outside the HLA region. This could be related to clinical observations that multiple autoimmune diseases in single subjects are more often seen in those with limited forms than those with diffuse-cutaneous forms. The PRS constructed from GWAS SNPs moderately fit the predictive model indicating a potential utility of PRS for a risk assessment of SSc in clinical settings. This was relevant to the preceding study, in which EUR-GWAS-based PRS had also shown moderate fitness with further improvement of the predictive performance by incorporating clinical laboratory parameters[57]. It is noteworthy that the threshold of GWAS p-value for the best predictive parameter sets was relatively small in both the previous and the present studies, suggesting the lower polygenicity of SSc compared to other autoimmune diseases, such as RA or SLE. Since laboratory test results were not available, especially for European subjects, the impact of including laboratory parameters on predictive performance should be investigated in future studies. On the other hand, prioritizing the top 5% SNPs annotated according to IRF8-binding in RAMOS cells improved the predictive performance showing the better trans-ancestral portability of PRS with the use of IMPACT-annotated SNPs.

The trans-ethnic GWAS meta-analysis identified a total of 30 GWAS loci, most of which were reproductions of the previous findings and derived from those identified in European populations. Nevertheless, three of these loci including *EOMES*, *ESR1*, and *SLC12A5* have never been reported, demonstrating the advantageous outcome of incorporating the Asian population. Eomesodermin encoded by *EOMES* and T-bet are TFs belonging to the T-box family and known to differently regulate CD8$^+$ T cell differentiation and function as well as exhaustion[58–60]. Considering the association of *EOMES* variation with multiple AIDs such as RA[61], multiple sclerosis[62], and ankylosing spondylitis[63], as well as immune-related traits including lymphocyte count and granulocyte count[64], variations of *EOMES* may be a trigger of autoimmunity shared among multiple AIDs including SSc. A nuclear hormone receptor, estrogen receptor 1, encoded by *ESR1* is involved in various gene expression which affects cellular proliferation and differentiation in given tissues. ESR1 binds to nuclear factor-κB (NF-κB) and these two molecules mutually trans-repress each other to regulate cellular response including cytokine production[65]. Of note, rs230534 in the *NFKB1* region was also one of the lead SNPs of our meta-analysis (Table 2). Since females are ~five times more affected by SSc than males, the association of *ESR1* with SSc is quite reasonable, but on the other hand, it was surprising that previous studies have never identified this locus; it might be due to the relatively weak association and effect sizes. Although rs9074 is positioned at 3'UTR of *SLC12A*, it was reported that rs9074 was eQTL and pQTL for CD40 expression. On the other hand, potassium-chloride transporter member 5 (KCC2) encoded by *SLC12A5* is a brain-specific chloride potassium symporter and is well-characterized in neuronal cells for its function of maintaining intracellular chloride concentrations[66]. Thus, it is likely that rs9074 modifies the disease susceptibility by affecting CD40 expression. We observed a high trans-ancestry genetic correlation between European and Japanese SSc (Supplementary Data 7), which was higher than the correlation between European and East Asian SLE (0.64, 95% CI 0.46 to

0.81)[67]. However, observed high standard error indicates less accurate estimate for SSc compared to SLE, and thus further validation with more sample size is mandatory in the future. It would also be interesting to test the with different populations such as African American or Latino.

As we expected, most of the susceptibility genes including novel risk loci identified in the present study overlapped with those associated with other AIDs, especially SLE. Consequently, there was a significant genetic correlation between SSc and SLE, and to a lesser extent RA despite the relatively small number of cases in the present study. Multi-trait analysis of GWAS (MTAG) jointly analyzes summary statistics from GWAS of different traits[68], and the recent multi-ancestry MTAG for SLE incorporating 10 genetically correlated AIDs successfully identified 16 novel loci[69]. It would be interesting to perform MTAG for both Japanese and European SSc to identify signals, that are associated with multiple AIDs and hence further support the concept of shared genetic architectures among multiple AIDs.

Taking together the findings of the present study, we can expand future studies to delineate clinicopathologic features of SSc. Despite the vigorous effort to pinpoint causal variants by fine-mapping or to predict functional consequences with various annotation tools, it is still unclear how the candidate causal variants affect the pathological process of SSc, such as progressive fibrosis or vasculopathy. Conventional stratification based on autoantibody profiles or distribution of skin lesions has been utilized and successfully characterized each clinical subtype with distinct genetic background in European[12]. The same approach was applied to East Asian SSc previously[11] as well as in the present study; however, these studies were still underpowered and hence need a larger sample size to identify the robust association. On the other hand, focusing on specific phenotypes, such as ILD, digital ulcers, pulmonary hypertension, or renal crisis, could also be a reasonable approach and provide information more relevant to specific pathologic processes of SSc beyond autoimmunity. As an example, a stronger association of the *TNFAIP3* SNPs with SSc-ILD than the whole SSc had been observed in Caucasians[20], and the same trend was also observed in the Japanese SSc of the current study. Integrating transcriptomics, epigenetics, and structural variations would further clarify the functional consequences of the causal variants while incorporating rare variants with the use of a WGS-based reference panel would enhance the accurate prediction of causal variants. In sum, clarifying fine phenotype-genotype association with functional annotation will help us understand the molecular basis of SSc pathology.

Our study did not answer several questions, such as how identified and unidentified susceptibility variants, HLAs, interaction with other genes[70], or environmental factors contribute to the difference in disease phenotypes among AIDs, and this question is also still open to be explored.

Nevertheless, our largest-scale Asian GWAS provides valuable insights into the complex genetic architecture of SSc as well as the prioritization of targets for future functional studies.

## Methods
### Ethics
This study was designed and conducted in compliance with all the relevant regulations regarding the use of human study participants and the criteria set by the Declaration of Helsinki, which was authorized by RIKEN Center for Integrative Medical Sciences. The study was also approved by the ethical committees at the RIKEN Center for Integrative Medical Sciences (17-17-16(16), 2020-40).

Written informed consent was obtained from all the participants before enrollment into the study, and no compensation was provided to the study participants for this study. This study was reviewed and approved by the individual institutional review boards of the following institutions.

RIKEN Center for Integrative Medical Sciences, Kyoto University, Tohoku University, The University of Tokyo, Chukyo Hospital, Nippon Medical School, Kyushu University, University of Fukui, Kanazawa University, University of Occupational and Environmental Health, Gunma University, Tokyo Women's Medical University, Tokai University, Kumamoto University, Sapporo Medical University, Nagoya City University, Fukushima Medical University, Osaka University, University of Tsukuba, Utano National Hospital, Hokkaido University, Toho University.

## Study subjects

A total of 1499 cases consisted of 712 who had been enrolled in the previous study[11] and 787 who were newly enrolled. A total of 112,609 controls consisted of 2105 from a single center and 110,504 from BBJ project in which 200,000 individuals with one of 47 common diseases were enrolled[16,17]. All the subjects were Japanese nations. The diagnosis of SSc was made by physicians according to the ACR/EULAR classification criteria (2013)[71]. The presence of ILD was judged per institution and based on the findings of roentgenograms and/or CT scan images.

Self-reporting gender and genotype-based sex were compared for sex determination, in which we found no discordance between them.

## Genotyping and quality control

All the case samples and the non-BBJ control samples were genotyped using Illumina Infinium CoreExome Array or a combination of Illumina Infinium Core Array and Illumina Infinium Exome Array. BBJ samples were genotyped using Illumina Human OmniExpress Exome BeadChip or a combination of Illumina HumanOmniExpress and HumanExome BeadChips (CoreExome Array and OmniExpress Exome BeadChip are not exome arrays, but genome-wide arrays).

The genotype QC criteria were set as follows and variants that did not meet any of these criteria were excluded from further analyses; genotyping call rate ≥99%, Hardy–Weinberg equilibrium $p$-values (HWE-P) $> 1.0 \times 10^{-6}$, allele frequency difference from those of the imputation panel <3.0%, and MAF > 0.01.

The subjects who met any of the following criteria were excluded from the analyses; subjects with genotyping call rate <0.98, those who were in a high degree of relatedness showing PiHAT > 0.25 estimated by PLINK1.9[72], or those who were outliers of East Asian (EAS) in the principal component analysis (PCA) (Supplementary Fig. 1, 2). For the PCA, we extracted variants shared between our datasets (Set1 and Set2) and those in the HapMap project.

## Imputation

After the sample and the variant QCs, genotypes were phased and imputed altogether to enhance the accuracy of genotyping imputation. For a reference genotype panel, we used a previously constructed imputation panel from the phase 3 1000 genome project ver.5 (1KGp3ver5) data (https://www.internationalgenome.org/data)[73] combined with high-depth WGS data from 3256 Japanese subjects from BBJ (J3K), which enables rare variant detection[74]. The genotype data were phased and imputed by EAGLE (ver.2.3.5) and Minimac4 (ver.1.0.0), respectively[75]. Variants with low imputation accuracy ($R^2 < 0.3$) or MAF < 0.01 among control samples were excluded. After the imputation, 9,246,028 autosomal variants and 255,517 X chromosome variants remained.

## Association analysis of Japanese SSc

An initial association analysis was conducted by a logistic regression model using PLINK (ver.2.0)[72]. Ten genetic principal components were used as covariates. For X chromosome analysis, males and females were separately analyzed first and then meta-analyzed with an inverse-variance fixed-effect model to estimate overall associations. The genome-wide significant threshold of $p$-value was set at $5.0 \times 10^{-8}$. Novel variants were defined as those that passed the genome-wide

significant threshold and had not been identified as significant variants in the previous GWASs or were at least 1 Mb apart from a significant locus in the previous GWASs. Conditional analyses were conducted using GCTA-COJO[76], where association analyses were repeated conditioning on given significant variants in a region within ±1 Mb from the variant until the association did not meet the threshold $p$-value of significant level. We used a relaxed threshold level, $p = 1.0 \times 10^{-6}$ [77], for the conditional analyses[78]. The obtained results were confirmed by conducting conditional analyses using PLINK2.0.

We also applied the Firth regression[18] using PLINK2.0 and a generalized linear mixed model using a scalable generalized linear mixed model for region-based association test (SAIGE, v1.1.6)[19] to address potential bias caused by low allele frequencies and case-control imbalance. For the Firth regression, all the case samples ($N = 1428$) were included while control samples consisted of randomly selected 13,946 samples from BBJ and the Set 1 control samples, which resulted in an improvement of case/control ratio from 1/79 to 1/11. For the linear mixed effect model, only identical samples were excluded in the kinship adjustment (PiHAT > 0.90).

## Fine-mapping analysis

In search of a causal SNP in each independently associated region, we utilized a statistical fine-mapping based on the asymptomatic Bayes factors. Bayes factor was calculated from minor allele frequencies and the $p$-values through Wakefield's approximations[79], and posterior probabilities that a single SNP is causal in the region based on the two key assumptions; (1) all SNPs have been genotyped, and (2) there is a single causal variant. The 95% credible set of SNPs located within 250 kb distance from the genomic position of each lead SNP was defined. For visualization, regional locus zoom plots were drawn with LocalZoom (v0.14.0)[80].

## Transcriptome-wide association study

TWAS, an association analysis for disease susceptibility based on gene expressions estimated by summary statistics, was conducted with FUSION software[81] using the GTEx ver.7 multi-tissue models consisting of 48 cell/tissue types[25]. We also conducted TWAS for subsets of white blood cells, CD4 T cells, CD8 T cells, NK cells, Monocytes, B cells, and neutrophils, using the Japanese eQTL study for 105 Japanese healthy subjects[26]. The statistical significance threshold was based on Bonferroni correction accounting for all the tested genes (0.05/279,331).

## Disease-related pathway analysis

Potential disease-related pathways were explored by FUMA (v1.3.8), an integrative online platform for functional mapping and annotation of genetic associations[82]. The GWAS summary statistics was uploaded for the initial SNP2GENE analysis and the resultant mapped genes were utilized for the GENE2FUNC analysis.

## Trans-ethnic genetic effect correlation

The transethnic genetic effect correlation between the European and Japanese populations was estimated using the python package, Popcorn (ver.0.9.9)[83]. Briefly, the cross-population scores were computed using the 1KGp3-EAS plus J3K and 1KGp3-EUR, and the heritability and the transethnic genetic correlation of a pair of GWAS summary statistics were fitted to the scores[73].

**Trans-ethnic meta-GWAS analysis with European meta-GWAS dataset.** A trans-ethnic meta-analysis between Japanese and European SSc was conducted by PLINK (ver.1.9) with the use of an inverse-variance fixed-effects model. The latest GWAS meta-analysis summary data of the European population consisted of 9095 cases and 17,584 controls[12] was used. Conditional analyses were also conducted using GCTA-COJO, where the association analysis was conducted by conditioning on significant variants in each population and the resultant

association data were meta-analyzed to obtain significant variants. The analyses were repeated until no variants reached a relaxed threshold level of significance, $p = 1.0 \times 10^{-6}$.

## SNP annotation

Functional annotations of given variants, including potential alteration of protein function for exonic variants, were identified by ANNOVAR (version: 2017-07-17)[84]. For the lead variants and their strong LD variants ($R^2 \geq 0.8$), we examined if these variants were also eQTL variants in the previous eQTL study of leukocyte subgroups[26] or GTEx ver.8[25]. Enrichment of GWAS significant variants in cell type-specific active histone marks was measured using Haploreg ver.4.1, an online tool exploring annotations for noncoding variants based on DNAse and chromatin immunoprecipitation (ChIP) sequencing data[29]. LoF variants or deleterious exonic variants were explored among the lead variants and their LD variants ($R^2 \geq 0.8$) by VEP/LOFTEE (v1.0.2) or Polyphen2 (v2.2.13) and SIFT (v5.2.2), respectively.

## Transcription factor binding motif analysis

Transcription factor binding motif analysis was conducted using Tomtom (v5.3.3)[85], an online-based motif comparison tool, using the cCRE sequence containing the reference allele (C) or the alternative allele (T) of rs10917688 as input sequences.

## LD estimation

The regional plots were drawn using LocusZoom software[86]. LD of a given variant with the corresponding lead variant was estimated using PLINK (ver.1.9) referring to the imputation results of the data of 1KGp3ver5 plus J3K.

## LD score regression

To estimate the heritability, the LDSC was conducted using LDSC software (ver.1.0.0)[87] with a liability scale under the disease prevalence of 0.1% for Japanese population[7] and 0.2% in European populations[1], respectively. Partitioned heritability enrichment was also measured in specific cell groups and detailed cell types using the baseline model (ver.2.2)[88]. To measure genetic correlations between SSc and various complex diseases[89], the summary statistics of 40 target diseases of BBJ and that of Japanese SLE[35] were utilized.

## gchromVAR

The gchromVAR weights chromatin features by posterior probabilities of fine-mapped variants and computes the enrichment for each cell type versus an empirical background matched for GC content and feature intensity[36]. The gchromVAR was applied to the Japanese SSc GWAS summary statistics using 18 bulk ATAC-seq for FACS-sorted hematopoietic progenitor populations derived from bone marrow samples of multiple healthy individuals. The threshold of significance was based on Bonferroni correction ($P = 0.002778$).

## Polygenic risk scores for SSc susceptibility

PRSs were calculated to assess the polygenic feature of SSc and the predictive ability of given SNPs for disease susceptibility following the online instruction (https://choishingwan.github.io/PRS-Tutorial/plink/).

Since the European dataset had the largest case numbers among the datasets used in the present study and thus enabled the most accurate approximate coefficients of variants, we initially used the summary statistics of the European GWAS to generate matrices consisting of $r^2$ of LD and GWAS p-values. To test the better approximation of the coefficients, we meta-analyzed the European and Set 1 Japanese GWASs and used the summary statistics.

Set 1 Japanese dataset was used as a discovery dataset to assess the predictive ability of PRS models and to determine the threshold of p-values for variants included in the construction of the PRS. Set 2

Japanese dataset was used as a test dataset to evaluate the performance of the PRS.

SNPs ($N = 3,652,217$) outside the HLA region shared between Japanese and European datasets were extracted and SNPs with a minor allele frequency <0.01 or those with an imputation quality score (Rsq) < 0.3 for each cohort were excluded. We applied the standard pruning and thresholding method to construct the PRS[90]. Specifically, the clump function of PLINK (1.90b) was used to generate eligible SNPs with the 250 kb window. A total of 9 pruning thresholds of LD ($r^2$) from 0.1 to 0.9 and a total of 20 thresholds of p-values in GWAS ($P_T$) ($5 \times 10^{-8}$, $5 \times 10^{-7}$, $1 \times 10^{-6}$, $5 \times 10^{-6}$, $1 \times 10^{-5}$, $5 \times 10^{-5}$, $5 \times 10^{-4}$, 0.05, 0.01, 0.05, 0.1, 0.2, 0.3, 0.4, 0.5, 0.6, 0.7, 0.8, 0.9, 1) were set to construct PRS matrices. For each, $r^2$ was used to generate different PRSs based on the $P_T$s. The natural logarithms of the GWAS odds ratios (ORs) were used as weights across all datasets. The SNP alleles used in the PRS were aligned to risk alleles for SSc susceptibility. All the effect sizes were set to positive values and the effect alleles were defined accordingly. The PRS was the sum of the weighted allele counts (by their respective GWAS effect sizes) across all the SNPs included in the PRS according to the following formula:

$$PRS_i = \sum_{k=1}^{n} \beta_k X_{k,i} \qquad (1)$$

where I denotes each subject, n denotes the number of variants passing a threshold, $\beta_k$ denotes an effect size of $k$-th SNP and $X_k,i$ denotes the genotype dosage of $k$-th SNP in individual i.

After the calculation of PRSs, a logistic regression model was applied to calculate the association between the PRS and susceptibility to SSc for the Japanese subjects. The area under the receiver operating characteristic (AUROC), sensitivity, and specificity of each model, was generated by the R package pROC version 1.13.0. To assess the goodness of fit of a given model, Nagelkerke's pseudo-$R^2$ metric was also calculated.

We split the dataset into 20 quantiles according to individual PRS to evaluate the performance of the PRS to distinguish case and control. The tenth quantile in 20 quantiles was used as the refe.[91] We applied logistic regression by comparing the remaining subgroups with the reference. For measuring correlation between PRS and age of disease onset, linear regression model was utilized and Spearman's correlation was calculated to evaluate the goodness of fit.

PRS and the accompanying statistics have been deposited and can be available at zenodo (https://doi.org/10.5281/zenodo.10152768).

## IMPACT

IMPACT (Inference and Modeling of Phenotype-related aCtive Transcription), is a genome annotation strategy to identify regulatory elements defined by cell-state-specific TF binding profiles and can capture more cis-eQTL variation than sequence-based annotations[37,38].

To examine the improvement of PRS performance with the use of IMPACT-annotated SNPs, the top 5% of SNPs annotated according to IRF8-binding in one of B cell lines, RAMOS cells, and the lead SNPs of the meta-analysis for the European and Set1 Japanese GWASs were prioritized to generate PRSs. The rest of the procedures were the same as described above, and the predictive performance and the goodness of fit of the model were compared with those obtained without prioritization of SNPs.

## Reporting summary

Further information on research design is available in the Nature Portfolio Reporting Summary linked to this article.

## Data availability

The GWAS summary statistics is available at figshare (https://doi.org/10.6084/m9.figshare.23823087). Individual-level data are protected

and not available to share on public repository in accordance to IRB-approved protocol, all requests to access these data can be made to the corresponding author: Chikashi Terao (chikashi.terao@riken.jp). The BioBank Japan data are available under the following accession codes: hum0311 [https://humandbs.biosciencedbc.jp/en/hum0311-v2] and hum0014 [https://humandbs.biosciencedbc.jp/en/hum0014-v26] upon request to the National Bioscience Database Center (NBDC) through the necessary application process (https://humandbs.biosciencedbc.jp/en/data-use). The 1000 Genome Project data utilized for imputation and LD calculation can be available at ISGR (https://www.internationalgenome.org/data). The latest European GWAS meta-analysis summary statistics is available at GWAS catalog under accession number 31672989 [https://www.ebi.ac.uk/gwas/publications/31672989]. Datasets utilized for TWAS are available at GTEx V7 (https://www.gtexportal.org/home/downloads/adult-gtex#qtl) and the NBDC Human Database (https://humandbs.biosciencedbc.jp/en/hum0099-v1). The latter dataset was also used for eQTL analysis together with ImmuNexUT dataset (https://www.immunexut.org) and Single-Tissue cis-QTL Data of GTEx V8 (https://www.gtexportal.org/home/downloads/adult-gtex#qtl). Overlap with candidate cis-regulatory elements are based on Regulome DB (https://regulomedb.org/regulome-search/), HaploReg (https://pubs.broadinstitute.org/mammals/haploreg/haploreg.php), and EONCODE (https://genome.ucsc.edu/cgi-bin/hgTracks?db=hg38&lastVirtModeType=default&lastVirtModeExtraState = &virtModeType=default&virtMode=0&nonVirtPosition = &position=chr1%3A161687350%2D161693349&hgsid=1774763328_zAe1Nkr1qIKljCBs4l6TTauio6qQ). Source data are provided with this paper. The source data file contains all the downstream analysis data except for polygenic scores, which are available at zenodo (https://doi.org/10.5281/zenodo.10152768).

## Code availability

For the codes for statistical analyses, we followed the publicly available codes and the instructions, which are provided by the following statistical tool website: PLINK1.9 (https://www.cog-genomics.org/plink/), PLINK2.0 (https://www.cog-genomics.org/plink/2.0/), SAIGE (https://github.com/weizhouUMICH/SAIGE), GCTA-COJO (https://yanglab.westlake.edu.cn/software/gcta/#COJO), ANNOVAR (https://annovar.openbioinformatics.org/en/latest/), Fine Mapping (Wakefield Method, https://rdrr.io/cran/gtx/man/abf.Wakefield.html), gchromVAR (https://github.com/caleblareau/gchromVAR), LDSC (https://github.com/bulik/ldsc), Popcorn (https://github.com/brielin/Popcorn), LocusZoom (http://locuszoom.org/), LocalZoom (https://statgen.github.io/localzoom/), The MEME Suite (https://meme-suite.org/meme/tools/meme) and HOMER (http://homer.ucsd.edu/homer/motif/), VEP/LOFTEE (https://registry.opendata.aws/hail-vep-pipeline/), SIFT (https://sift.bii.a-star.edu.sg/www/publications.html), PolyPhen-2 (http://genetics.bwh.harvard.edu/pph2/index.shtml, FUMA (https://fuma.ctglab.nl/), Polygenic Risk Score Analyses (https://choishingwan.github.io/PRS-Tutorial/), IMPACT (https://github.com/immunogenomics/IMPACT). No custom codes were generated for the present study.

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

## Acknowledgements

We thank all the individuals who participated in the study and the staff in the BBJ project for their efforts. We also thank Midori Suzuki in Aichi Cancer Center Research Institute for the array genotyping experiments and all members in the Laboratory for Statistical and Translational Genetics, RIKEN Center for Integrative Medical Sciences for the technical support. This study was supported by Medical Research and Development (AMED) under Grant Number JP21ek0109555, JP21tm0424220, JP21ck0106642, JP23ek0410114, and JP23tm0424225, Japan Society for the Promotion of Science KAKENHI Grant JP20H00462, THE KATO MEMORIAL TRUST FOR NAMBYO RESEARCH and the Medical Research Support Project of the Shizuoka Prefectural Hospital Organization. CT received all the grants listed above. NT, NO, and HS were supported by the RIKEN Junior Research Associate Program.

## Author contributions

Conceptualization: Y.I., C.T. Methodology: Y.I., N.T., N.O., H.S., Y.K., K.T., S.Y., X.L., S.I., K.H., T.A., C.T. Investigation: Y.I., N.T., C.T. Visualization: Y.I., C.T. Funding acquisition: C.T. Project administration: C.T. Supervision: C.T. Writing – original draft: Y.I., C.T. Writing – review & editing: Y.I., N.T., Y.A., M.K., Y.S., M.A., M.H., T.M., K.S., S.M., H.Y., A.Y., T.K., K.T., A.O., M.K., Y.T., Y.I., K.N., H.K., A.U., A.S., H.N., M.J., K.M., T.M., H.I., M.Y., C.S., H.T., E.N., A.M., T.Y., M.F., Y.K., D.G., T.S., N.A., H.Y., T.H., T.A., H.E., Y.S., A.K., J.H., N.O., H.S., Y.K., K.T., S.Y., X.L., S.I., K.H., A.S., Y.M., S.I., Y.T., O.I., K.T., T.T., S.S., Y.O., T.M., F.M., K.M., T.A., I.I., K.M., M.K., Y.K., K.O., C.T.

## Competing interests

The authors declare no competing interests.

## Additional information

Yuki Ishikawa [1], Nao Tanaka [1,2], Yoshihide Asano[3,4], Masanari Kodera[5], Yuichiro Shirai[6], Mitsuteru Akahoshi [7,8], Minoru Hasegawa[9], Takashi Matsushita [10], Kazuyoshi Saito[11], Sei-ichiro Motegi[12], Hajime Yoshifuji[13], Ayumi Yoshizaki[4], Tomohiro Kohmoto[14], Kae Takagi[15], Akira Oka [16], Miho Kanda[5], Yoshihito Tanaka[5], Yumi Ito[5], Kazuhisa Nakano[11],

Hiroshi Kasamatsu[9], Akira Utsunomiya[9], Akiko Sekiguchi[12], Hiroaki Niiro[7], Masatoshi Jinnin[17], Katsunari Makino[18], Takamitsu Makino[18], Hironobu Ihn[18], Motohisa Yamamoto[19], Chisako Suzuki[20], Hiroki Takahashi[20], Emi Nishida[21,22], Akimichi Morita[21], Toshiyuki Yamamoto[23], Manabu Fujimoto[24], Yuya Kondo[25], Daisuke Goto[25], Takayuki Sumida[25], Naho Ayuzawa[26], Hidetoshi Yanagida[26], Tetsuya Horita[27], Tatsuya Atsumi[27], Hirahito Endo[28], Yoshihito Shima[29], Atsushi Kumanogoh[29], Jun Hirata[30], Nao Otomo[1], Hiroyuki Suetsugu[1], Yoshinao Koike[1], Kohei Tomizuka[1], Soichiro Yoshino[1], Xiaoxi Liu[1], Shuji Ito[1], Keiko Hikino[31], Akari Suzuki[32], Yukihide Momozawa[33], Shiro Ikegawa[34], Yoshiya Tanaka[11], Osamu Ishikawa[12], Kazuhiko Takehara[10], Takeshi Torii[35], Shinichi Sato[4], Yukinori Okada[30], Tsuneyo Mimori[13,36], Fumihiko Matsuda[37], Koichi Matsuda[38,39], Tiffany Amariuta[40,41,42,43,44], Issei Imoto[45], Keitaro Matsuo[46], Masataka Kuwana[6], Yasushi Kawaguchi[47], Koichiro Ohmura[13] & Chikashi Terao[1,48,49] ✉

[1]RIKEN Center for Integrative Medical Sciences, The Laboratory for Statistical and Translational Genetics, Yokohama, Japan. [2]Department of Rheumatology, Graduate School of Medical and Dental Sciences, Tokyo Medical and Dental University, Tokyo, Japan. [3]Department of Dermatology, Tohoku University Graduate School of Medicine, Sendai, Japan. [4]Department of Dermatology, The University of Tokyo, Tokyo, Japan. [5]Department of Dermatology, Chukyo Hospital, Japan Community Health Care Organization, Nagoya, Japan. [6]Department of Allergy and Rheumatology, Nippon Medical School Graduate School of Medicine, Tokyo, Japan. [7]Department of Medicine and Biosystemic Science, Kyushu University Graduate School of Medical Sciences, Fukuoka, Japan. [8]Department of Rheumatology, Saga University Hospital, Saga, Japan. [9]Faculty of Medical Sciences, Department of Dermatology, University of Fukui, Fukui, Japan. [10]Department of Dermatology, Faculty of Medicine, Institute of Medical, Pharmaceutical and Health Sciences, Kanazawa University, Kanazawa, Japan. [11]The First Department of Internal Medicine, University of Occupational and Environmental Health, Japan, Kitakyushu, Japan. [12]Department of Dermatology, Gunma University Graduate School of Medicine, Maebashi, Japan. [13]Department of Rheumatology and Clinical Immunology, Graduate School of Medicine, Kyoto University, Kyoto, Japan. [14]Aichi Cancer Center Research Institute, Division of Molecular Genetics, Nagoya, Japan. [15]Tokyo Women's Medical University, Adachi Medical Center, Tokyo, Japan. [16]Department of Molecular Life Sciences, Division of Basic Medical Science and Molecular Medicine, Tokai University School of Medicine, Isehara, Japan. [17]Department of Dermatology, Wakayama Medical University Graduate School of Medicine, Wakayama, Japan. [18]Department of Dermatology and Plastic Surgery, Faculty of Life Sciences, Kumamoto University, Kumamoto, Japan. [19]Department of Rheumatology and Allergy, IMSUT Hospital, The Institute of Medical Science, The University of Tokyo, Tokyo, Japan. [20]Department of Rheumatology and Clinical Immunology, Sapporo Medical University School of Medicine, Sapporo, Japan. [21]Department of Geriatric and Environmental Dermatology, Nagoya City University Graduate School of Medical Sciences, Nagoya, Japan. [22]Department of Dermatology, Okazaki City Hospital, Okazaki, Japan. [23]Department of Dermatology, Fukushima Medical University, School of Medicine, Fukushima, Japan. [24]Department of Dermatology, Graduate School of Medicine, Osaka University, Osaka, Japan. [25]Department of Rheumatology, Institute of Medicine, University of Tsukuba, Tsukuba, Japan. [26]Department of Clinical Immunology, National Hospital Organization, Utano National Hospital, Kyoto, Japan. [27]Faculty of Medicine and Graduate School of Medicine, Department of Rheumatology, Endocrinology and Nephrology, Hokkaido University, Sapporo, Japan. [28]Omori Medical Center, Toho University, Rheumatic Disease Center, Tokyo, Japan. [29]Department of Respiratory Medicine and Clinical Immunology, Osaka University Graduate School of Medicine, Osaka, Japan. [30]Immunology Frontier Center, Osaka University, Statistical Immunology, Osaka, Japan. [31]RIKEN Center for Integrative Medical Sciences, The Laboratory for Pharmacogenomics, Yokohama, Japan. [32]RIKEN Center for Integrative Medical Sciences, The Laboratory for Autoimmune Diseases, Yokohama, Japan. [33]RIKEN Center for Integrative Medical Sciences, The Laboratory for Genotyping Development, Yokohama, Japan. [34]RIKEN Center for Integrative Medical Sciences, The Laboratory for Bone and Joint Diseases, Yokohama, Japan. [35]Torii Clinic, Maizuru, Japan. [36]Ijinkai Takeda General Hospital, Kyoto, Japan. [37]Graduate School of Medicine, Kyoto University, Center for Genomic Medicine, Kyoto, Japan. [38]Institute of Medical Science, The University of Tokyo, Laboratory of Genome Technology, Human Genome Center, Tokyo, Japan. [39]Department of Computational Biology and Medical Sciences, Laboratory of Clinical Genome Sequencing, Graduate School of Frontier Sciences, The University of Tokyo, Tokyo, Japan. [40]Center for Data Sciences, Harvard Medical School, Boston, MA, USA. [41]Divisions of Genetics and Rheumatology, Department of Medicine, Brigham and Women's Hospital, Harvard Medical School, Boston, MA, USA. [42]Program in Medical and Population Genetics, Broad Institute of MIT and Harvard, Cambridge, MA, USA. [43]Department of Biomedical Informatics, Harvard Medical School, Boston, MA, USA. [44]Graduate School of Arts and Sciences, Harvard University, Cambridge, MA, USA. [45]Aichi Cancer Center Research Institute, Nagoya, Japan. [46]Aichi Cancer Center Research Institute, Division of Cancer Epidemiology and Prevention, Nagoya, Japan. [47]Tokyo Women's Medical University, Division of Rheumatology, Department of Internal Medicine, Tokyo, Japan. [48]Shizuoka General Hospital, The Clinical Research Center, Shizuoka, Japan. [49]The Department of Applied Genetics, School of Pharmaceutical Sciences, University of Shizuoka, Shizuoka, Japan. ✉e-mail: chikashi.terao@riken.jp

