## [Peer Review File · Nature Communications]

GWAS for systemic sclerosis identified six novel susceptibility loci including one in the Fcγ receptor regionReviewer #1 (Remarks to the Author):

The authors are to be commended for this substantial contribution to the understanding of the genetic contribution to SSc susceptibility in the Japanese population and in the trans-ethnic genetic analysis combining Japanese and European-derived populations. The inclusion of the large number of controls (from a participating site and from the Biobank of Japan, BBJ) clearly strengthens the analysis.

There are many strengths of this work, most notable of which is the finding of three novel regions that are associated with SSc susceptibility. The inclusion of the trans-ethnic meta-GWAS analysis is of value.

However, clarification of the following points would be helpful to the reader:

1. The abstract states that the study identified 3 novel signals but then only mentions two (FCGR/FCRL region and IRF8) – this should be clarified; also there should be clarification in the Abstract regarding signals identified in the Japanese subjects only versus those found in the trans-ethnic meta-GWAS.

2. The terminology of systemic and limited types of SSc is confusing. SSc is a systemic disease both in its limited and in its diffuse cutaneous forms, including centromere antibody positive disease. Therefore, a more precise terminology would be welcome – in line 299 of the paper in the section entitled “Different genetic architectures between the systemic and the limited types of SSc” – suggest “Different genetic architecture among the major clinical and serological subtypes” which you then proceed to describe. Also delete the term “systemic” from lines 303, 305, 314 and multiple other subsequent places in the text.

3. How was ILD defined? If defined per the institutional criteria (or similar language), this is acceptable but it needs to be stated.

4. Reference #7 (F.C. Arnett et al) noted on line 34 -please note – this was not a GWAS but an analysis confined to the HLA region.

Reviewer #2 (Remarks to the Author):

The manuscript by Ishikawa et al. is a genetic association study of severe rheumatic disease, systemic sclerosis, with analysis of 1,428 patients and >100,000 controls from Japanese population. Additionally, trans ethnic analysis together with 9,095 patients from European population and >17,000 controls was performed. These analyses were focused on non-HLA variations and reveal several new association hits and confirmed previously known associations with better confidence. This is an original study that is highly valuable and can be used as a reference for the future genetic studies of systemic sclerosis. The conclusions well follow what was found by researchers and I did not find any faults that may prevent this study from the publication. The availability of the summary statistics for population of interest will facilitate further developments in this area and is very helpful. The study is based on well-established methodology and corresponds to current state of art in this area.

My only concern is regarding the interpretation of the data from the HLA locus. It is a difficult task to analyze the associations within this locus and it deserves a separate publication. In the current study, the major focus is on non-HLA variations and the part of the manuscript that gives very limited information regarding the association of the HLA variations with the subgroups of SyS appeared as a complementary and not fully elaborated. It looks like the association signals from the HLA are by far more strong and outrun non-HLA associations. This part (Figure 5) does not align with the rest of the text. My suggestion will be to remove it from the manuscript and to explain in the

Discussion session that non-HLA variations were studied with intention to extend it to HLA in a future.

Reviewer #3 (Remarks to the Author):

Ishikawa et al conducted a GWAS in systemic sclerosis by expanding their previous studies through an increase in the number of disease affected patients and mostly by a much larger numbers of controls. The patient population was split in 2 sets for methodological reasons but most of the important results only appear in combined analyses. A new variant in FCGR region with a potential functional relation with IRF8 has been identified. FCGR variant was enriched in histone marked on B cells. Then a meta GWAS with the largest EU one is performed raising new association and showing a genomic risk score with AUC at 0.59.

The work is well done with a relevant progression and complementary experiments to try to provide insight regarding the new variants. One limitation may be the lack of biological pathways that could be related to the findings. One may see the work as the addition of new statistically associated variants but without new clues regarding the disease pathogenesis or targets for biomarkers or drug.

In the introduction section, the authors comment the available knowledge but chose on purpose data from GWAS or candidate gene studies. One may suggest to mostly discuss and comment on the most robust data that were mainly provided by the large EU meta-GWAS (Nat Commun. 2019 Oct 31;10:4955°.

By example, although coming from this group the past PLD4 association coming from a candidate gene approach seems less robust. This loci might be part of the discussion section but could be disregarded in the introduction section. Providing separately Asian data and their commonalities or specificities is of interest and fits with the overall aim of the herein study. It would be of interest to comment about the SSc Asian, or SSc Japanese, phenotype because in case of more severe disease, one may anticipate more relevant genetic data.

Regarding the SSc patients, there is no specific table to describe them. When stratification on subsets is provided we can see expected proportions of diffuse or limited cutaneous patients. Expected propositions are seen for auto-antibodies. One is missing that is RNAPolIII that is also very specific, could the authors comment on this. One group could also be of interest that is the one without autoantibodies or with anti-nuclear antibodies but without SSc specificities. In a context of seeking B-cell role, these sub-groups could be of interest. Lung involvement became in the recent series the major prognostic factor; we learn herein that 625 patients could have ILD among the whole set, which is relevant. However, how was it defined? Were all patients phenotyped for lung? It is important because by example TNFAIP3 association were stronger in SSc patients with lung disease.

Regarding phenotyping, the authors highlight that several genes contribute to multiple AID; it is known that SSc phenotype can be enriched by polyautoimmunity and this may weight on GWAS results. Could multi-trait analysis across autoimmune disorders of significant genetic correlation with SSc using MTAG in each ancestry separately and combined the results across ancestries using fixed effect meta-analysis be run ?(Nature Communications 2023;14:668).

For the novel SNPs derived from the Japanese population, the separation in 2 sets seems artificial and the main results come from pooled samples. It is anticipated that some control populations were genotyped using different genotyping platforms, and maybe between cases and controls, how this could have impacted the results?

Because functional impact beyond genome location is of critical importance, as this was done in lupus, could Open Targets Genetics database be used? (Nature Communications

2023;14:668).

Effects of ancestry can be important. Recently in systemic lupus the trans-ancestral genetic-effect correlation between the Chinese and European GWAS was estimated to be 0.64 (CI 0.46 to 0.81) (Nature Communications 2021; 12: 772). In SSc, herein, the trans-ethnic genetic correlation estimate is 0.738 ± 0.418 . Can the authors comment on the respective results and their meanings?

The detailed analyses of FCGR are fine and relationships with IRF8 is stimulating. Some data about IRF8 in SSc were previously reported and should be commented herein (J Invest Dermatol 2021 Aug;141(8):1954-1963 and Arthritis Research & Therapy 2015;17,71)

For the different genetic architectures between the "systemic" and the "limited" types of SSc, there is a misunderstanding of the authors about the disease substratification. Both limited cutaneous and diffuse cutaneous are "systemic" diseases and it does not make sense to compare "limited" with "systemic". The subsetting is done according to skin extent. By example ILD can occur in both subsets and this part should be revised accordingly.

The part about the polygenic risk score is interesting but sounds as of limited impact. First of all, a comparison with the previous study looking at PRS should be discussed (Ann Rheum Dis 2021 Jan;80(1):118-127). In lupus, integrating PRS with conventional lab tests led to further improvement in diagnostic accuracy, could this be performed in SSc. Early diagnosis is also a challenge and there is growing literature about early disease and pre-scleroderma, revised PRS could be of help.

Predictive performances were higher for lcSSc or ACA-positive SSc but this might be explained because there is probably a higher immune component in these subsets (other autoimmune diseases are mostly seen in LcSSc patients) and because the PRS contains mainly immune genes, association could be stronger in LcSSc.

The discussion should be improved. The text about FCR is quite long and what applies to SSc is vague. What is expected is any new clue for relevant new biological effects or shared data from other diseases with clinical impact. The reviewer would suggest to rather try to focus with SSc and position any new clue with what is known in SSc pathogenesis and what should be done to give sense of the new finding in a disease like SSc where, beyond autoimmunity, vasculopathy and fibrosis drive the prognosis.

Moreover, the authors miss some of previous findings and do not comment them. One single example, among several, is the TNFAIP3 variants that is identified. By example, the Japanese population provides a signal at 50290949 whereas in the EU population 5029939 was identified. Could the authors expand on this variants, LD, populations, fine mapping, functional effects...

We deeply appreciate all the helpful comments from the reviewers as well as the time and efforts they made for the reviewing process. Herein, we present our point-by-point responses to all the comments. All the changes we made in the text are highlighted in red. We hope that all the responses are satisfactory, leading to a great improvement of the manuscript.

Reviewer #1 (Remarks to the Author):

The authors are to be commended for this substantial contribution to the understanding of the genetic contribution to SSc susceptibility in the Japanese population and in the trans-ethnic genetic analysis combining Japanese and European-derived populations. The inclusion of the large number of controls (from a participating site and from the Biobank of Japan, BBJ) clearly strengthens the analysis.

There are many strengths of this work, most notable of which is the finding of three novel regions that are associated with SSc susceptibility. The inclusion of the trans-ethnic meta-GWAS analysis is of value.

We appreciate the understanding of the value of our work by the reviewer.

However, clarification of the following points would be helpful to the reader:

1. The abstract states that the study identified 3 novel signals but then only mentions two (FCGR/FCRL region and IRF8) – this should be clarified; also there should be clarification in the Abstract regarding signals identified in the Japanese subjects only versus those found in the trans-ethnic meta-GWAS.

Thank you for pointing these out. We included the motivation to focus on *FCGR/FCRL* loci (lines 100-102) and a description about novel significant loci in trans-ancestry meta-analysis and in the Japanese GWAS (lines 106-107).

We think that these amendments will help readers understand the significant signal in the *FCGR/FCRL* region was only one specific to Japanese-SSc and thus we focused on the signal for further downstream analyses.

2. The terminology of systemic and limited types of SSc is confusing. SSc is a systemic disease both in its limited and in its diffuse cutaneous forms, including centromere antibody positive disease. Therefore, a more precise terminology would be welcome – in line 299 of the paper in the section entitled “Different genetic architectures between the systemic and the limited types of SSc” – suggest “Different genetic architecture among the major clinical and serological subtypes” which you then proceed to describe. Also delete the term “systemic” from lines 303, 305, 314 and multiple other subsequent places in the text.

Thank you very much for your helpful comment. We fully agree to the point the reviewer raised and replaced “systemic” with “diffuse (forms).” Accordingly, the section title was also edited, which is more appropriate than the original one (lines 319-320).

3. How was ILD defined? If defined per the institutional criteria (or similar language), this is acceptable but it needs to be stated.

This is an important point missing in the manuscript initially submitted. The presence of ILD was judged per institution and based on the findings of roentgenograms and/or CT scan images. This information was added to the corresponding part of the Methods, lines 589-590.

4. Reference #7 (F.C. Arnett et al) noted on line 34 -please note – this was not a GWAS but an analysis confined to the HLA region.

We agree with the reviewer and amended the corresponding part of the Introduction, lines 140-142, including the references cited there.

Reviewer #2 (Remarks to the Author):

The manuscript by Ishikawa et al. is a genetic association study of severe rheumatic disease, systemic sclerosis, with analysis of 1,428 patients and >100,000 controls from Japanese population. Additionally, trans ethnic analysis together with 9,095 patients from European population and >17,000 controls was performed. These analyses were focused on non-HLA variations and reveal several new association hits and confirmed previously known associations with better confidence. This is an original study that is highly valuable and can be used as a reference for the future genetic studies of systemic sclerosis. The conclusions well follow what was found by researchers and I did not find any faults that may prevent this study from the publication. The availability of the summary statistics for population of interest will facilitate further developments in this area and is very helpful. The study is based on well-established methodology and corresponds to current state of art in this area.

We greatly appreciated the hard work of the reviewer to interpret our work. As the reviewer kindly suggested, we are going to deposit the summary statistics after the publication in order to facilitate further studies.

My only concern is regarding the interpretation of the data from the HLA locus. It is a difficult task to analyze the associations within this locus and it deserves a separate publication. In the current study, the major focus is on non-HLA variations and the part of the manuscript that gives very limited information regarding the association of the HLA variations with the subgroups of SyS appeared as a complementary and not fully elaborated. It looks like the association signals from the HLA are by far

more strong and outrun non-HLA associations. This part (Figure 5) does not align with the rest of the text. My suggestion will be to remove it from the manuscript and to explain in the Discussion section that non-HLA variations were studied with intention to extend it to HLA in a future.

Thank you very much for having a thoughtful comment. We totally agree that the impact of the HLA variations could be examined separately from those of the non-HLA variants. We also thought that the observed different non-HLA associations among clinical subtypes would be of interest to people in the corresponding field. Accordingly, we amended this section so as not to highlight the different associations of the HLA locus, but to focus on the SNPs outside the HLA region (lines 319-343). We also mentioned a potential future study for the HLA association in the Discussion (lines 410-412).

Reviewer #3 (Remarks to the Author):

Ishikawa et al conducted a GWAS in systemic sclerosis by expanding their previous studies through an increase in the number of disease affected patients and mostly by a much larger numbers of controls. The patient population was split in 2 sets for methodological reasons but most of the important results only appear in combined analyses. A new variant in FCGR region with a potential functional relation with IRF8 has been identified. FCGR variant was enriched in histone marked on B cells. Then a meta GWAS with the largest EU one is performed raising new association and showing a genomic risk score with AUC at 0.59.

The work is well done with a relevant progression and complementary experiments to try to provide insight regarding the new variants. One limitation may be the lack of biological pathways that could be related to the findings. One may see the work as the addition of new statistically associated variants but without new clues regarding the disease pathogenesis or targets for biomarkers or drug.

Thank you very much for the supportive and constructive comments on our work. We believe that we could further add deep insight into the pathogenesis of SSc as well as potential biomarkers or drug targets through this revision process.

In the introduction section, the authors comment the available knowledge but chose on purpose data from GWAS or candidate gene studies. One may suggest to mostly discuss and comment on the most robust data that were mainly provided by the large EU meta-GWAS (Nat Commun. 2019 Oct 31;10:4955°. By example, although coming from this group the past PLD4 association coming from a candidate gene approach seems less robust. This loci might be part of the discussion section but could be disregarded in the introduction section.

We strongly agree that the latest EUR meta-GWAS (Lopez-Izac, E., et al. Nat Commun, 2019) provided by far the most robust genetic association data in Europeans and contributed to our deeper understanding of the genetic architecture of SSc. On the other hand, that of East Asian SSc have been lacking for so long and we have been wondering whether the underlying genetic

architectures are different between Europeans and East Asians, and, if so, how they are different. Such examples include *TNFAIP* and *PLD4* loci, which for the first time fulfilled the genome-wide significant level ($p < 5 \times 10^{-8}$) by increasing statistical power in the current study (driven by East Asians). Although the associations were weaker in European SSc than in Japanese SSc (Table 2) and thus less robust compared to other variants in Europeans, such population-specific signals would also enhance our understanding of SSc pathology, which is one of the aims of the present study. Accordingly, we think it is important to put an emphasis on the less robust association in the previous studies focusing on East Asian SSc referring to the well-established European data. On the other hand, as the reviewer suggested, we moved the description of the variants with less robust associations in the previous studies, those of *TNFAIP3* and *PLD4*, to the Discussion (lines 413-422).

Providing separately Asian data and their commonalities or specificities is of interest and fits with the overall aim of the herein study. It would be of interest to comment about the SSc Asian, or SSc Japanese, phenotype because in case of more severe disease, one may anticipate more relevant genetic data.

Thank you very much for another helpful comment. It was reported that SSc in Asians is characterized by the younger age of onset, higher frequency of diffuse skin involvement, higher frequency of ATA and anti-U1 RNP antibodies, and more severe clinical phenotype leading to poorer prognosis compared to non-Asian SSc (PMID 35585950). As the reviewer commented, the phenotypic difference from non-Asian SSc might be attributed to the underlying genetic architecture and hence one of the strong motivations of the current study. We added the characteristic features of Asian SSc (lines 133-137).

On the other hand, only three genome-wide approaches have been conducted in East Asian SSc so far. Due to the very limited sample sizes, none of the loci except for that of the *STAT4* region were convincing and thus both East-Asian specific signals or those shared between EAS and EUR have never been identified. We previously conducted the GWAS for clinical subtypes of SSc, but, again, none of the loci outside the HLA region were identified to be significantly associated. Taking the limitation of these studies into consideration, we also supplemented the description of the East Asian studies in the Introduction (lines 149-161).

Regarding the SSc patients, there is no specific table to describe them.

When stratification on subsets is provided we can see expected proportions of diffuse or limited cutaneous patients. Expected propositions are seen for auto-antibodies. One is missing that is RNAPolIII that is also very specific, could the authors comment on this. One group could also be of interest that is the one without autoantibodies or with anti-nuclear antibodies but without SSc specificities.

Thank you very much for pointing an important point out. Demographic features are provided in the updated Supplementary Table S14.

For anti-RNAP-III antibodies, we don't have enough data to estimate the frequency among Japanese SSc as well as to run the association test. Considering the lower frequency than those of ATA and ACA among SSc patients (PMID 35382018), a larger sample size is mandatory, and this is one of the unmet needs to be investigated in future studies. We put an additional sentence to the Discussion part.

In a context of seeking B-cell role, these sub-groups could be of interest.

We totally agree that finer cell-type specificity could provide more insightful information. We refer to the ImmuneNext database, eQTL database of 28 immune cell types including B cell subtypes obtained from Japanese subjects with 10 immune-mediated diseases including SSc and healthy subjects (Ota, et al. Cell, 2021, DOI: <https://doi.org/10.1016/j.cell.2021.03.056>). We found relatively specific expression of FCGR2B in B cells, among which unswitched memory (USM) B cells expressed the highest level. SSc patients-derived USM B cells as well as other B cell subsets expressed less FCGR2B than those of healthy subjects. Another perfect LD variant, rs10917698, was identified as a significant eQTL for FCGR2B in plasmablasts. We added this information (lines 303-310), which further strengthens the evidence of the rolls of B cells in SSc.

Lung involvement became in the recent series the major prognostic factor; we learn herein that 625 patients could have ILD among the whole set, which is relevant. However, how was it defined? Were all patients phenotyped for lung? It is important because by example TNFAIP3 association were stronger in SSc patients with lung disease.

This is an important point missing in the manuscript initially submitted. The presence of ILD was judged per institution and based on the findings of roentgenograms and/or CT scan images. This information was added to the corresponding part of the Methods, lines 589-590.

For TNFAIP3 variations, we confirmed the stronger effect in SSc subjects with ILD as the reviewer implied, showing the relevancy of our cohorts. This finding was also incorporated in the section for clinical subtypes (lines 337-341, Supplementary Table S17).

	MAF (case control)		OR	95 % CI		P-value
whole SSc vs Ctrl	0.102	0.070	1.50	1.32	1.69	1.7E-10
SSc w/ ILD vs Ctrl	0.108	0.070	1.59	1.33	1.91	4.0E-07
SSc w/o ILD vs Ctrl	0.094	0.070	1.37	1.11	1.68	0.0029
SSc w/ ILD vs SSc w/o ILD	0.108	0.094	1.18	0.89	1.56	0.2395

Regarding phenotyping, the authors highlight that several genes contribute to multiple AID; it is known that SSc phenotype can be enriched by polyautoimmunity and this may weight on GWAS results. Could multi-trait analysis across autoimmune disorders of significant genetic correlation with SSc using MTAG in each ancestry separately and combined the results across ancestries using fixed effect meta-analysis be run?(Nature Communications 2023;14:668).

Thank you very much for the insightful comment. We have run MTAG by incorporating the publicly available summary statistics of SLE and RA because these two AIDs had shown significant or near-significant genetic correlations with SSc in the present study. Accordingly, we have identified more than twice as many lead variants in both Japanese and European SSc. Among the loci newly identified by the MTAG, many of them were those which have been identified for associations with multiple AIDs, such as *PTPN22*, *CD28-CTLA4*, *IL2-IL21*, *WDFY4*, or *ITGAM*. Furthermore, those identified only in the EUR-SSc were significantly associated with JPN-SSc and vice versa. Since we are preparing a separate manuscript for the MTAG results, we have decided not to mention the results in detail, but briefly highlighted the results to further support the genetic architecture shared among multiple AIDs in the Discussion (lines 549-555).

For the novel SNPs derived from the Japanese population, the separation in 2 sets seems artificial and the main results come from pooled samples. It is anticipated that some control populations were genotyped using different genotyping platforms, and maybe between cases and controls, how this could have impacted the results?

Thank you very much for your comments. Please note that we did not artificially split the data sets into two. Since the Set1 dataset had been utilized in our previous study (PMID 28314753), we independently analyzed the newly enrolled Set 2 dataset, which is a well-accepted approach in GWAS papers.

The cases in the two datasets were genotyped with the same array chipset but at different time points. Set 1 controls were genotyped as the same platform as Set 1 cases and we did not observe substantial inflation in the association test for the Set 1 dataset (λ_{GC} 1.075, Supplementary Figure 4).

On the other hand, Set 2 controls were those from the BBJ samples and had been genotyped with the different genotyping chips as described in the Methods. Nonetheless, no substantial inflation was observed (λ_{GC} 1.038) in the association test.

Taken together, we believe that the difference in genotyping chip had not impacted our study results.

Because functional impact beyond genome location is of critical importance, as this was done in lupus, could Open Targets Genetics database be used? (Nature Communications 2023;14:668).

Thank you very much for the helpful suggestion. We refer to Open Targets Genetics for further functional annotations of the novel lead variants identified in the Japanese GWAS as well as the trans-ancestry meta-analysis. We found that the *FCGR/FCRL* variants, rs6697139 and rs10917688, were both pQTL for FCGR2A and 2B expression in plasma and added the information in the main text (lines 314-315).

For the novel risk variants of the trans-ancestry meta-analysis, we added the eQTL and pQTL information of rs398390, an intergenic variant between *LINC01980-CMC1-EOMES* (lines 245-246). Also, we amended the description of rs9074, which is positioned at 3'UTR of *SLC12A* but is more

likely to affect CD40 expression both at transcript and protein levels (lines 245-246, lines 535-539).

Effects of ancestry can be important. Recently in systemic lupus the trans-ancestral genetic-effect correlation between the Chinese and European GWAS was estimated to be 0.64 (CI 0.46 to 0.81) (Nature Communications 2021; 12: 772). In SSc, herein, the trans-ethnic genetic correlation estimate is 0.738 ± 0.418 . Can the authors comment on the respective results and their meanings?

Thank you very much for pointing that out. The high trans-ancestry genetic correlations indicate that the underlying genetic architectures are similar between two populations, Europeans and East Asians, for those two autoimmune diseases, SLE and SSc. However, the observed high SE (Supplementary Table 7) also indicated that less accurate estimation for SSc, and thus further validation with more sample size is mandatory for a more accurate estimation. Testing the other populations, such as African American or Latino, will also be of interest and hence is remained for future studies. We add the comments in the Discussion (lines 540-545).

The detailed analyses of FCGR are fine and relationships with IRF8 is stimulating. Some data about IRF8 in SSc were previously reported and should be commented herein (J Invest Dermatol 2021 Aug;141(8):1954-1963 and Arthritis Research & Therapy 2015;17,71)

Thank you very much for the helpful suggestion. We incorporated these important references in the Discussion (lines 475-482).

For the different genetic architectures between the "systemic" and the "limited" types of SSc, there is a misunderstanding of the authors about the disease substratification. Both limited cutaneous and diffuse cutaneous are "systemic" diseases and it does not make sense to compare "limited" with "systemic". The subsetting is done according to skin extent. By example ILD can occur in both subsets and this part should be revised accordingly.

Thank you very much for your helpful comment. We are very sorry for our careless mistakes. We strongly agree to the point the reviewer raised and replaced "systemic" with "diffuse (forms)."

The part about the polygenic risk score is interesting but sounds as of limited impact. First of all, a comparison with the previous study looking at PRS should be discussed (Ann Rheum Dis 2021 Jan;80(1):118-127). In lupus, integrating PRS with conventional lab tests led to further improvement in diagnostic accuracy, could this be performed in SSc. Early diagnosis is also a challenge and there is growing literature about early disease and pre-scleroderma, revised PRS could be of help.

We strongly agree that we should have described the preceding PRS study in the paper above. Since we excluded the variants in the HLA region due to different associations between EUR and

JPN, we could not directly compare the performance of our study with the previous one. Nevertheless, considering the numbers of SNPs included in the best predictive parameter set were relatively small, and the numbers of SNPs which fulfilled the threshold GWAS p-value were also relatively small both in the previous study and ours, polygenicity of SSc might be weaker than other autoimmune diseases, such as RA or SLE. This is also supported by the relatively low heritability estimates (h^2).

Unfortunately, laboratory data were unavailable in many subjects, especially in the EUR meta-analysis. However, as shown in the papers above, combining basic laboratory test results with PRS likely to show a better predictive performance, which could be useful in clinical settings, and hence needs further investigation in future collaborative studies.

We described the preceding study in comparison with ours and the potential utility of PRS combined with basic laboratory parameters in the Discussion (lines 507-514).

Predictive performances were higher for lcSSc or ACA-positive SSc but this might be explained because there is probably a higher immune component in these subsets (other autoimmune diseases are mostly seen in LcSSc patients) and because the PRS contains mainly immune genes, association could be stronger in LcSSc.

Thank you very much for the insightful comment. As we presented, dcSSc or ATA subsets had shown a stronger association with the HLA genes than lcSSc or ACA. In addition, the limited forms of SSc were associated with genes other than HLA genes, while diffuse forms were solely driven by the HLA genes. Together these data support the idea that the contribution of multiple genes including immune-related genes (outside the HLA region) to the pathogenesis of lcSSc is relatively higher than that of dcSSc. We added these comments in the Discussion (lines 501-505).

The discussion should be improved. The text about FCR is quite long and what applies to SSc is vague. What is expected is any new clue for relevant new biological effects or shared data from other diseases with clinical impact. The reviewer would suggest to rather try to focus with SSc and position any new clue with what is known in SSc pathogenesis and what should be done to give sense of the new finding in a disease like SSc where, beyond autoimmunity, vasculopathy and fibrosis drive the prognosis.

Thank you very much for your helpful comment. As the reviewer suggested, we cut the description of *FCGR/FCRL* variants to a minimum (lines 434-461) and add how to incorporate the findings of the present study together with the previous evidence into future research focusing more on the pathologic process of SSc, such as progressive fibrosis or vasculopathy (lines 556-574).

Moreover, the authors miss some of previous findings and do not comment them. One single example, among several, is the TNFAIP3 variants that is identified. By example, the Japanese population provides a signal at 50290949 whereas in the EU population 5029939 was identified. Could the authors expand on this variants, LD, populations, fine mapping, functional effects...

Thank you very much for referring to another important point. Having comments on the previous findings by comparing them to the present study is of special importance, especially in the Discussion, and hence we added brief comments that the current study is valid enough to reproduce the previous findings (lines 412-413). As seen in the example of *TNFAIP3* locus, it is not rare that different variants are tagged with true causal variants between different populations mainly due to different LD structures. Indeed, rs5029949 and rs5029939 are in almost perfect LD in European (D' 1.0, r^2 0.9148). In such a case, it is natural to consider that true causal variants are common between the populations and try to fine-map the causal variants. This approach worked well and we successfully narrowed down the number of causal variants in various loci including the *TNFAIP3* locus (Supplementary Table 9). The top-ranked intronic variant, rs5029937, is marked with active histone marks, such as H3K4me, H3K27ac, or H3K4me3, in T cells and B cells as well as one of lung fibroblast cell lines (IMR90) and skin fibroblasts (Haploreg v4.2). Similar findings were also observed for other fine-mapped SNPs, which have been known as the risk SNPs without functional annotations. We updated this information in the manuscript (lines 264-268, lines 417-418, Supplementary Table S9).

Reviewer #1 (Remarks to the Author):

The revised manuscript has been significantly clarified and adds an important contribution to the understanding of genetic susceptibility factors in SSc patients.

Reviewer #2 (Remarks to the Author):

The authors responded to the criticism and I have no more comments regarding this manuscript.

Reviewer #3 (Remarks to the Author):

I think that the revised ms has been improved and I thank the authors for including the points that I raised